# Beneficial impacts of neuromuscular electrical stimulation on muscle structure and function in the zebrafish model of Duchenne muscular dystrophy

Elisabeth A Kilroy[1], Amanda C Ignacz[1], Kaylee L Brann[2], Claire E Schaffer[2], Devon Varney[2], Sarah S Alrowaished[2], Kodey J Silknitter[1], Jordan N Miner[3], Ahmed Almaghasilah[1], Tashawna L Spellen[2], Alexandra D Lewis[2], Karissa Tilbury[1,3], Benjamin L King[1,4], Joshua B Kelley[1,4], Clarissa A Henry[1,2]*

[1]Graduate School of Biomedical Science and Engineering, University of Maine, Orono, United States; [2]School of Biology and Ecology, University of Maine, Orono, United States; [3]Department of Chemical and Biomedical Engineering, University of Maine, Orono, United States; [4]Department of Molecular and Biomedical Sciences, University of Maine, Orono, United States

**Abstract** Neuromuscular electrical stimulation (NMES) allows activation of muscle fibers in the absence of voluntary force generation. NMES could have the potential to promote muscle homeostasis in the context of muscle disease, but the impacts of NMES on diseased muscle are not well understood. We used the zebrafish Duchenne muscular dystrophy (*dmd*) mutant and a longitudinal design to elucidate the consequences of NMES on muscle health. We designed four neuromuscular stimulation paradigms loosely based on weightlifting regimens. Each paradigm differentially affected neuromuscular structure, function, and survival. Only endurance neuromuscular stimulation (eNMES) improved all outcome measures. We found that eNMES improves muscle and neuromuscular junction morphology, swimming, and survival. Heme oxygenase and integrin alpha7 are required for eNMES-mediated improvement. Our data indicate that neuromuscular stimulation can be beneficial, suggesting that the right type of activity may benefit patients with muscle disease.

*For correspondence: clarissa.henry@maine.edu

## Editor's evaluation

This is an interesting and well-conceived study that explores the potential benefit of electrical stimulation for muscular dystrophy in terms of muscle structure and motor function. The authors take advantage of the zebrafish model system, and a well-characterized zebrafish mutant that models Duchenne muscular dystrophy, to show that certain stimulation paradigms can improve muscle morphology and muscle performance, like via integrin-mediated pathway(s). The potential implications of this research are broad as they begin to address the key question in the MD field about whether and what types of exercise may (or may not) be beneficial to dystrophic muscle.

## Introduction

Skeletal muscle is a dynamic tissue whose structural and molecular networks change in response to demand. Skeletal muscle's ability to adapt is critical to maintaining not only muscle health, but also the overall health of an individual. Skeletal muscle is one of the primary predictors of longevity and recovery from illness and injury, demonstrating that robust skeletal muscle mass is essential for

whole-body homeostasis (*Margolis and Rivas, 2015*). Whereas a great deal is known about the structural and functional plasticity of healthy skeletal muscle, far less is understood about plasticity and adaptation in diseased muscle. Muscular dystrophies are debilitating progressive diseases without cures. Here, we describe how neuromuscular electrical stimulation (NMES) impacts the zebrafish model of Duchenne muscular dystrophy (DMD).

Individuals with DMD harbor mutations in the gene encoding the protein dystrophin (*Hoffman et al., 1987*). Dystrophin provides a link between the actin cytoskeleton and the extracellular matrix (ECM) and serves as a scaffold for the assembly of the dystrophin-glycoprotein complex (DGC) within the sarcolemma (muscle plasma membrane) (*Bonilla et al., 1988*; *Ervasti and Campbell, 1991*). The stability and integrity of the ECM and DGC are critical to the viability of muscle fibers during contraction. The ECM modulates mechanical homeostasis and cell-matrix interactions (*Grzelkowska-Kowalczyk, 2016*; *Humphrey et al., 2014*). ECM-mediated distribution and transmission of force across muscle fibers are mostly achieved through the ECM-cytoskeleton linkage via the DGC (*Humphrey et al., 2014*; *Ramaswamy et al., 2011*). The DGC maintains the structural integrity of the sarcolemma and serves as a scaffold for various signaling and channel proteins as well as an anchoring point for signaling molecules near their sites of action (*Constantin, 2014*). The progressive muscle wasting and weakness in DMD is thought to result at least in part because the lack of dystrophin and the disruption of the DGC mechanically weaken the sarcolemma. Stress placed on a mechanically weakened sarcolemma causes microlesions to develop along the sarcolemma, increasing calcium entry, which results in muscle protein degradation and muscle fiber necrosis (*Alderton and Steinhardt, 2000*; *Gailly, 2002*; *Gillis, 1996*; *Ruegg et al., 2002*). Given the role that the DGC plays in muscle during force generation and the concern about potentially increased muscle damage, there has been interest in studying the effects of activity on the progression of muscle disease in humans. However, studies tend to be small with heterogeneous populations and a clear answer has not yet been established (*Alemdaroğlu et al., 2015*; *Bushby et al., 2010*; *Gianola et al., 2013*; *Hyzewicz et al., 2015*; *Jansen et al., 2013*; *Markert et al., 2012*). The lack of a clear answer is because these studies vary with regards to exercise regimes, controls (contralateral limbs versus different subjects), time frames of treatments, outcome measures, and conclusions. For example, one study conducted in 1966 concluded that a resistance exercise program is most effective at increasing strength if instituted early in the disease, and a study conducted in 1981 concluded that resistance exercise negatively affects walking (*Scott et al., 1981*; *Vignos and Watkins, 1966*). More recent studies have not examined resistance training but do suggest that bicycle ergometers can positively affect the quality of life early in the disease course (*Alemdaroğlu et al., 2015*; *Jansen et al., 2013*). Unfortunately, studies using the *mdx* mouse to study the impact of exercise on disease progression also take different approaches – treadmill training, voluntary wheel running, swimming, rotarod training – beginning at 3–96 weeks, leading to disparate results. Collectively, studies demonstrate either improvements in twitch tension, decreased necrosis, antioxidant capacities, and oxidative enzyme activity (*Baltgalvis et al., 2012*; *Call et al., 2008*; *Hulmi et al., 2013*); or reduced strength, increased edema and inflammation, increases in reactive oxygen species, and increases in lipid peroxidation (*Burdi et al., 2009*; *Faist et al., 2001*; *Kobayashi et al., 2012*). Most of these studies measured specific outcomes on individual muscles of the hindlimb, diaphragm, or heart. These muscles are not equally affected by the absence of dystrophin (*Louboutin et al., 1993*) and thus likely to respond differently to exercise. Taken together, these data clearly indicate that the impact of exercise on the progression of DMD is not very well characterized.

Skeletal muscle fibers are influenced by the activity pattern imposed upon them, whether the activity is from the innervating neuron or electrical stimulation (*Pette and Vrbová, 1985*). Early researchers, including Guillaume Benjamin Amand Duchenne, the French neurologist who described DMD in 1861, proposed that super-imposing electrical stimulation on dystrophin-deficient muscles could serve as a potential therapy (*Barnard et al., 1986*; *Duchenne, 1870*; *Reichmann et al., 1981*). NMES delivers a series of waveforms of electrical current that is characterized by its frequency, amplitude, and pulse width (or pulse duration) (*Sheffler and Chae, 2007*). These three parameters dictate the strength of the muscle contraction and the amount of force that is generated. The main advantage of NMES is its ability to activate fast- and slow-twitch muscle fibers, resulting in hypertrophy without high-effort voluntary force generation (*Gondin et al., 2011*). It is possible that NMES could impact muscle deterioration in congenital muscular dystrophies. However, only a few studies have examined

the impact of NMES on muscle strength and function in humans (*Scott et al., 1990*; *Scott et al., 1986*; *Zupan, 1992*; *Zupan et al., 1993*) and the *mdx* mouse model of DMD (*Dangain and Vrbova, 1989*; *Luthert et al., 1980*; *Vrbová and Ward, 1981*). The only clear conclusion from these studies is that NMES does not appear to be detrimental.

Zebrafish are an attractive model to elucidate the consequences of NMES on diseased muscle. Many molecular, ultrastructural, and histological features are shared between zebrafish and human muscle, including components of the DGC, the excitation-contraction coupling machinery, and the contractile apparatus (*Dou et al., 2008*; *Dowling et al., 2009*; *Guyon et al., 2003*; *Parsons et al., 2002*). Further, zebrafish exhibit reproducible, quantitative motor behaviors beginning at 1 day post-fertilization (dpf) (*Saint-Amant and Drapeau, 1998*), providing simple and noninvasive measures of muscle function. Dystrophin-deficient zebrafish, known as sapje[ta222a/ta222a], and referred here as *dmd* mutants, are the smallest vertebrate model of DMD. These *dmd* mutants exhibit severe structural and functional deficits by 4 dpf, and die prematurely between their second and third weeks (*Bassett et al., 2003*; *Berger et al., 2010*).

The purpose of this study was to use a longitudinal study design to evaluate the impacts of NMES on disease progression in *dmd* mutant zebrafish. Different NMES programs that varied in pulse frequency and voltage had differing effects on muscle structure, neuromuscular junction (NMJ) structure, motility, and life span. We did identify one program that improved all of the above. This program also increased the resilience of muscle fibers to high-force contraction, increased sarcomere length, and improved nuclear morphology. Deep sequencing indicated that, at least 3 days after exercise, transcriptional changes likely did not drive the improvement. However, deep sequencing did identify a couple of candidate mechanisms that were tested further. Heme oxygenase signaling and integrin alpha7 (Itga7) play roles in endurance NMES (eNMES)-mediated improvement. Taken together, our results show that NMES can have a dramatic positive impact on *dmd* muscle and establish the zebrafish as an excellent model for longitudinal studies that are critical for elucidating the basic mechanisms of skeletal muscle plasticity.

## Results

### A model for studying the impact of NMES on *dmd* muscle

Strength training is an excellent approach to combat muscle wasting and weakness in healthy individuals. Using zebrafish larvae as a model for lifting weights is not feasible, so we asked whether we could use NMES as an alternate means to stimulate muscle activity and combat muscle wasting and weakness in *dmd* larvae. Previous work used NMES as a controlled stimulus designed to illuminate the fact that muscle that appeared morphologically normal was highly susceptible to stimulation-induced injury (*Subramanian and Schilling, 2014*). We approached NMES from the opposite perspective and asked whether different NMES paradigms would have different impacts on neuromuscular stability in *dmd* larvae. The parameters that can be adjusted in NMES include pulse frequency and voltage. We designed four different NMES paradigms ranging from high-frequency/low-voltage pulse trains to lower-frequency/higher-voltage pulse trains (*Figure 1B and C*). In order to easily differentiate these paradigms from each other, and because they were conceptually based on strength training paradigms that vary in the number of repetitions and load, we named these paradigms endurance NMES (eNMES), hypertrophy NMES (hNMES), strength NMES (sNMES), and power NMES (pNMES). We first asked whether these different NMES paradigms elicited unique tail bend patterns that vary in how many times the tail bends as well as how hard it bends. As would be expected, eNMES with high-frequency/low-voltage pulse trains elicited a fast but subtle tail beat. Conversely, with pNMES, the tail beat infrequently but bent to a much greater degree (*Video 1*).

### NMES does not result in immediate damage to the sarcolemma

One of the major reasons why strength training is not recommended for individuals with DMD is due to the fragility of the sarcolemma and its susceptibility to contraction-induced damage. Prior to evaluating each NMES paradigm, we asked whether the pulse parameters resulted in dramatic immediate damage to the sarcolemma. We did this by asking whether increased Evans blue dye (EBD) was observed in muscle after one session of NMES. EBD was injected into the pericardial space at 2 dpf and allowed to circulate for 4 hr. Then, images of EBD in the zebrafish trunk musculature were

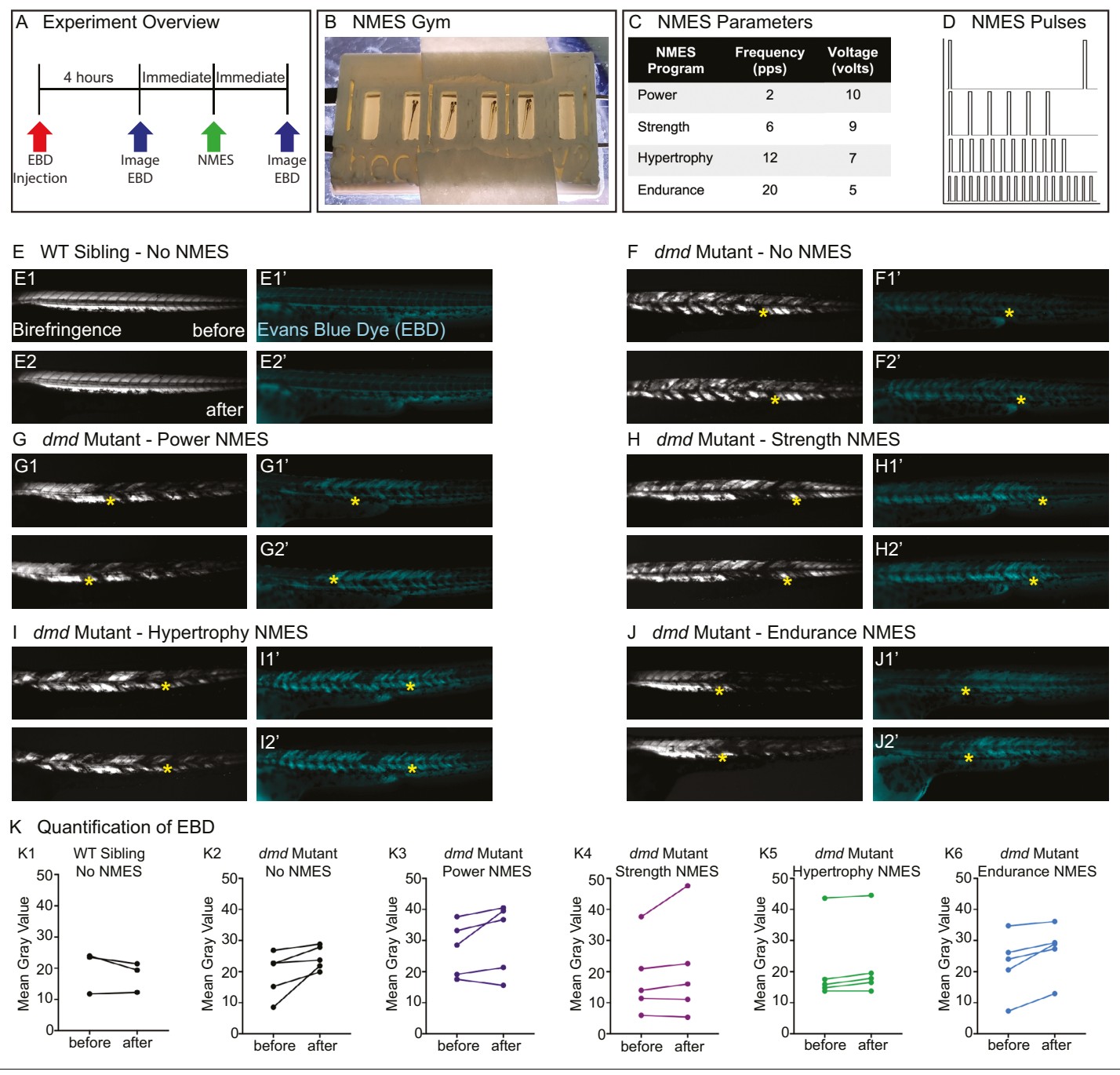

**Figure 1.** Four neuromuscular electrical stimulation (NMES) paradigms do not result in immediate damage to the sarcolemma. (**A**) Experimental overview. At 2 days post-fertilization (dpf), WT siblings and *dmd* mutants were injected with Evans blue dye (EBD). 4 hr later, zebrafish were imaged for birefringence and EBD before and after a single session of NMES. (**B**) For NMES, zebrafish are placed in a 3D-printed gym with their heads towards the positive electrode and tails towards the negative electrode. (**C, D**) NMES delivers a series of square wave pulses that vary in frequency and voltage. We named these paradigms after weightlifting regimes. (**E–J**) Anterior left, dorsal top, side-mounted birefringence, and EBD fluorescent images. Yellow asterisks denote the same position in embryos before and after NMES. (**E**) WT sibling control exhibits healthy muscle segments (**E1, E2**) and no dye entry in the muscle (**E1', E2'**) during the first and second imaging sessions. (**F**) *dmd* mutant control has significant areas of degenerated muscle (**F1**) and dye entry (**F1'**) but no new areas of degeneration or dye entry during the second imaging session (**F2, F2'**). (**G–J**) Similar to the *dmd* mutant control, *dmd* mutants that receive NMES have significant areas of degenerated muscle and dye entry prior to NMES but no new areas of degeneration or dye entry during following NMES. (**K**) Quantification of EBD during the first and second imaging sessions.

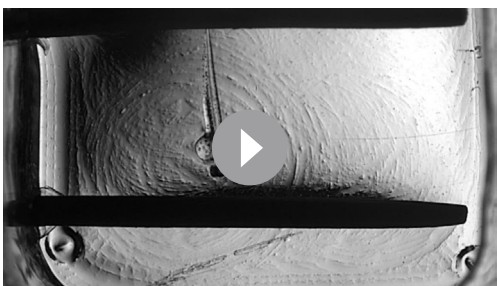

**Video 1.** Video of embryos undergoing different electrostimulation paradigms. The paradigms are noted in the video.
https://elifesciences.org/articles/62760/figures#video1

taken immediately prior to and after one session of NMES (*Figure 1A*). The relative amount of EBD in muscle was calculated using mean gray values of the EBD channel prior to and after NMES. Birefringence and EBD images of the same embryos before and after NMES are shown in *Figure 1*. The yellow stars denote the same position in the embryo before and after stimulation. Both wild-type (WT) and *dmd* mutant larvae are similar when imaged prior to and after NMES (*Figure 1E–J*). None of the NMES paradigms consistently caused a dramatic change in either birefringence (not shown) or EBD infiltration (*Figure 1K*, n = 5 embryos imaged, subjected to NMES, and imaged). These results indicated that the four NMES paradigms did not cause imme-diate dramatic damage to the sarcolemma.

## Different NMES paradigms differentially impact *dmd* muscle structure, function, and survival

After determining that NMES paradigms did not cause immediate dramatic damage, we asked whether different NMES programs had different effects on the progression of muscle degeneration in *dmd* larvae. We developed a protocol that was divided into two periods: the training period and the recovery period (*Figure 2A*). During the training period, zebrafish completed three sessions of NMES, each session lasting 1 min, on three consecutive days (2, 3, and 4 dpf) at the same time each day. Following these three training days, zebrafish entered the recovery period (5, 6, 7, and 8 dpf). The only aspects that changed across these experiments were the NMES pulse parameters. We used a longitudinal study design to elucidate the response of individual larvae to these different NMES paradigms.

WT larvae with all four NMES paradigms were unaffected (*Figure 2B1, C1, D1, E1*, and data not shown). There is some variability in the *dmd* phenotype. Thus, two examples of control and treated larvae are shown for each condition with all larvae quantified in panels labeled 6. For this figure, red arrowheads denote degeneration from the previous time point and green arrowheads denote improvement (either regeneration or hypertrophy) from the previous time point. We focused on the change in mean gray value from 5 dpf to 8 dpf because that change represents how the muscle responds to and recovers from three sessions of NMES. Between days 5 and 8, birefringence levels for control larvae trend towards slight improvement (*Figure 2B6, C6, D6, E6*). The eNMES and pNMES paradigms improved muscle structure in *dmd* mutants. pNMES resulted in a slight but significant increase in birefringence compared to controls (n = 32 control, n = 36 pNMES; p=0.0462; two biological replicates) (*Figure 2B6*, note also green arrows in *Figure 2B4, B5*). eNMES also increased regeneration between 5 and 8 dpf (n = 66 control, n = 66 eNMES; p=0.0037; five biological replicates) (*Figure 2E6*, note green arrows in *Figure 2E4, E5*). In contrast, *dmd* mutants that underwent sNMES exhibited significantly lower changes in mean gray values compared to control *dmd* mutants (n = 49 control, n = 47 sNMES; p=0.0302; two biological replicates) (*Figure 2C6*) while hNMES trended towards lowering birefringence (n = 38 control, n = 48 hNMES; p=0.1322; two biological replicates) (*Figure 2D6*). These data indicate that different NMES paradigms do have different effects on muscle structure in zebrafish larvae.

Birefringence provides a gross overview of muscle structure. Phalloidin stains filamentous actin and provides a finer assessment of muscle structure. Muscle fibers in WT zebrafish are highly organized and linear (*Figure 3A*). In contrast, many fibers in *dmd* mutants are thinner, wavy, and disorganized (*Figure 3B*, red arrows). Some fibers are also clearly degenerated and detached from their surrounding matrix (*Figure 3C1, D1, E1*, red arrowheads). We quantified the percentage of muscle segments with fiber degeneration. Neither sNMES nor hNMES affected the frequency of muscle fiber degeneration (*Figure 3D2, E2*). However, similar to the results observed with birefringence, eNMES (n = 22 control, n = 18 eNMES; p=0.0191; one biological replicate) and pNMES (n = 12 control, n = 20 pNMES; p=0.0033; one biological replicate) resulted in fewer fibers degenerating compared

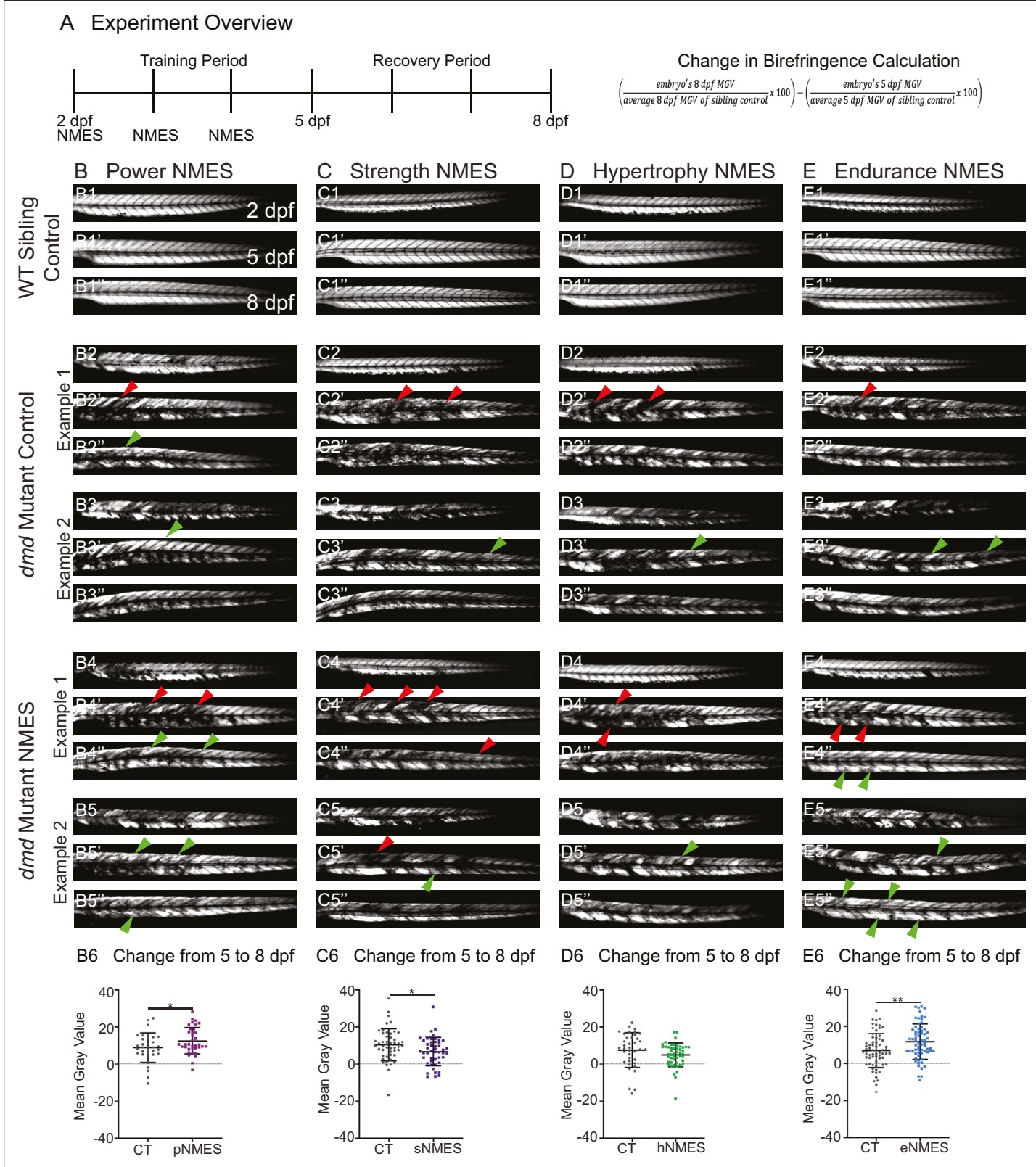

**A** Experiment Overview

**B** Power NMES

**C** Strength NMES

**D** Hypertrophy NMES

**E** Endurance NMES

Change in Birefringence Calculation

$$\left(\frac{embryo's\ 8\ dpf\ MGV}{average\ 8\ dpf\ MGV\ of\ sibling\ control} x\ 100\right) - \left(\frac{embryo's\ 5\ dpf\ MGV}{average\ 5\ dpf\ MGV\ of\ sibling\ control} x\ 100\right)$$

**B6** Change from 5 to 8 dpf

**C6** Change from 5 to 8 dpf

**D6** Change from 5 to 8 dpf

**E6** Change from 5 to 8 dpf

**Figure 2.** Impacts of neuromuscular electrical stimulation (NMES) paradigms on muscle structure through time. (**A**) Experimental overview and calculation of change in mean gray value from 5 to 8 days post-fertilization (dpf). At 2 dpf, birefringence images were taken followed by the first session of NMES. At 3 and 4 dpf, zebrafish underwent the second and third sessions of NMES, respectively. Birefringence images were taken at 5 and 8 dpf. The training program was divided into the training period (2–4 dpf) and the recovery period (5–8 dpf). (**B–E**) Anterior left, dorsal top, side-mounted

*Figure 2 continued on next page*

*Figure 2 continued*

birefringence images for WT sibling controls (panels labeled 1), control *dmd* mutants (two examples shown, panels labeled 2 and 3), and NMES-treated *dmd* mutants (two examples shown, panels labeled 4 and 5). The NMES regimens are labeled as such: panels labeled **B** were treated with power NMES (pNMES), panels labeled **C** were treated with strength NMES (sNMES), panels labeled **D** were treated with hypertrophy NMES (hNMES), and panels labeled **E** were treated with endurance NMES (eNMES). The change in mean gray values from 5 dpf to 8 dpf represents how the muscle responds to and recovers from three sessions of NMES and is shown in panels labeled 6. Positive changes indicate improvements in muscle structure while negative changes indicate deterioration in muscle structure. Red arrowheads denote degeneration from the previous time point, green arrowheads denote regeneration from the previous time point. pNMES (**B6**, maroon squares) and eNMES (**E6**, blue squares) significantly improve muscle structure in *dmd* mutants compared to *dmd* mutant controls (gray circles). sNMES (**C6**, purple squares) significantly worsens muscle structure in *dmd* mutants while hNMES (**D6**, green squares) trends to decrease muscle structure compared to *dmd* mutant controls (gray circles). Each data point represents a single zebrafish. Birefringence data were analyzed using two-sided *t*-tests. *p<0.05, **p<0.01.

to control *dmd* mutants (*Figure 3C2, F2*). Taken together, the above data indicate that eNMES and pNMES improve muscle structure in *dmd* larvae.

The NMJ is altered in DMD patients and animal models for DMD (*Ng and Ljubicic, 2020*). To our knowledge, NMJ morphology has not been investigated in the zebrafish model for *dmd*. We thus strove to characterize NMJs in *dmd* larvae and then asked whether NMJ morphology changed with NMES. We analyzed NMJ morphology by using the SV2 antibody to label presynaptic structures and

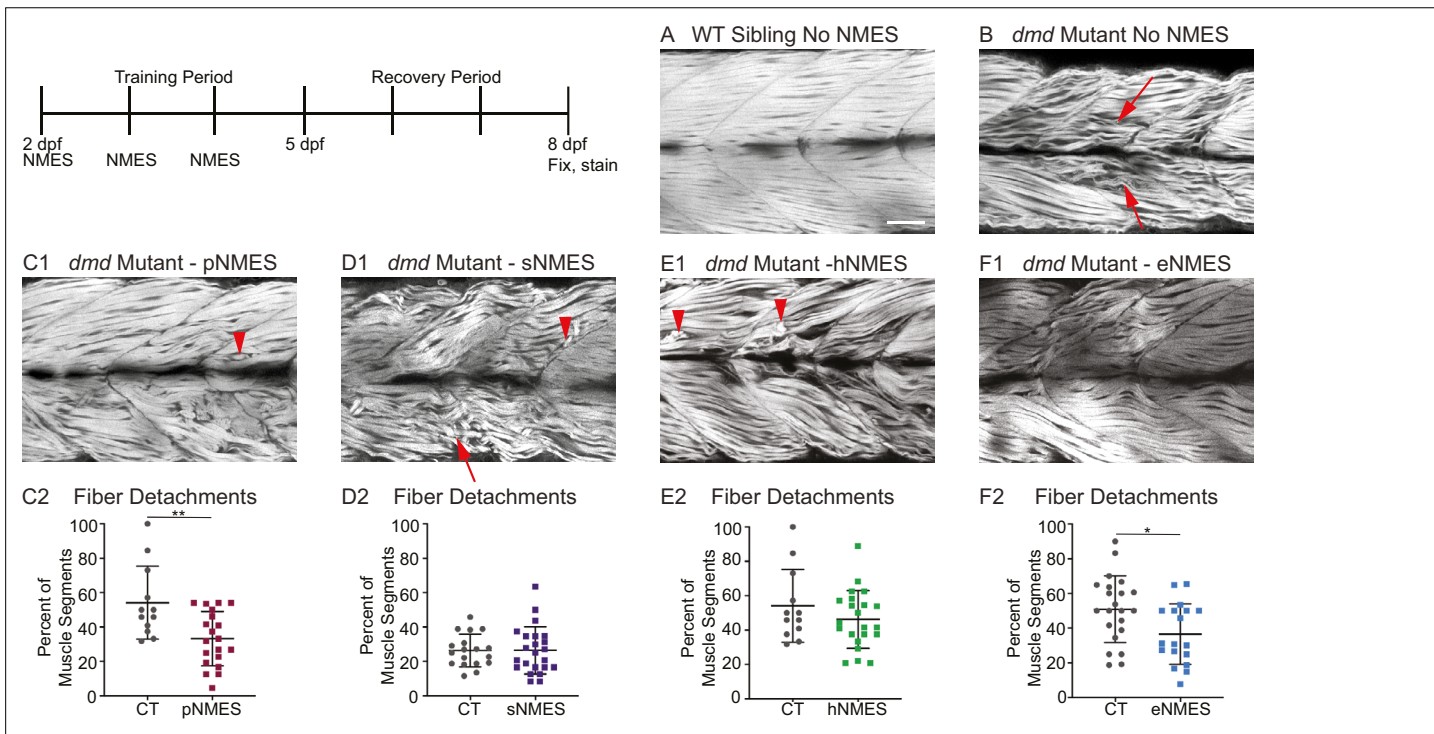

**Figure 3.** Impacts of neuromuscular electrical stimulation (NMES) paradigms on muscle fiber structure and degeneration. Phalloidin staining for F-actin at 8 days post-fertilization (dpf) allows for visualization of individual muscle fibers and the ability to count detached fibers in *dmd* mutants. Anterior left, dorsal top, side mounted. Red arrows point to disorganized muscle fibers, and red arrowheads point to detached muscle fibers. (**A**) Representative image of WT sibling demonstrates organized muscle fibers with well-defined myotome boundaries. (**B**) Representative image of *dmd* mutants demonstrates disorganized, wavy muscle fibers with poorly defined myotome boundaries and empty space between individual muscle fibers. (**C1**) Representative image of *dmd* mutant that received power NMES (pNMES) demonstrates less muscle fiber waviness, lack of empty space between muscle fibers but visible detached fibers. (**D1**) Representative image of *dmd* mutant that received strength NMES (sNMES) demonstrates massive deterioration of muscle fiber structure, disorganized myotomes with poorly defined boundaries. (**E1**) Representative image of *dmd* mutant that received hypertrophy NMES (hNMES) demonstrates improved muscle fiber organization with more defined myotome boundaries but visibly detached muscle fibers and empty space between fibers. (**F1**) Representative image of *dmd* mutant that received endurance NMES (eNMES) demonstrates healthy myotomes with clearly defined boundaries, organized muscle fibers with very few wavy fibers, and lack of empty space between fibers. Quantification of the percentage of muscle segments with detachments indicates that pNMES (**C2**) and eNMES (**F2**) significantly reduce fiber detachments in *dmd* mutants. sNMES (**D2**) and hNMES (**E2**) do not impact the percent of muscle segments with detachments. Each data point represents a single fish. A muscle segment is defined as half of a myotome. Muscle detachment data were analyzed using two-sided *t*-tests. *p<0.05, **p<0.01. Scale bar is 50 μm.

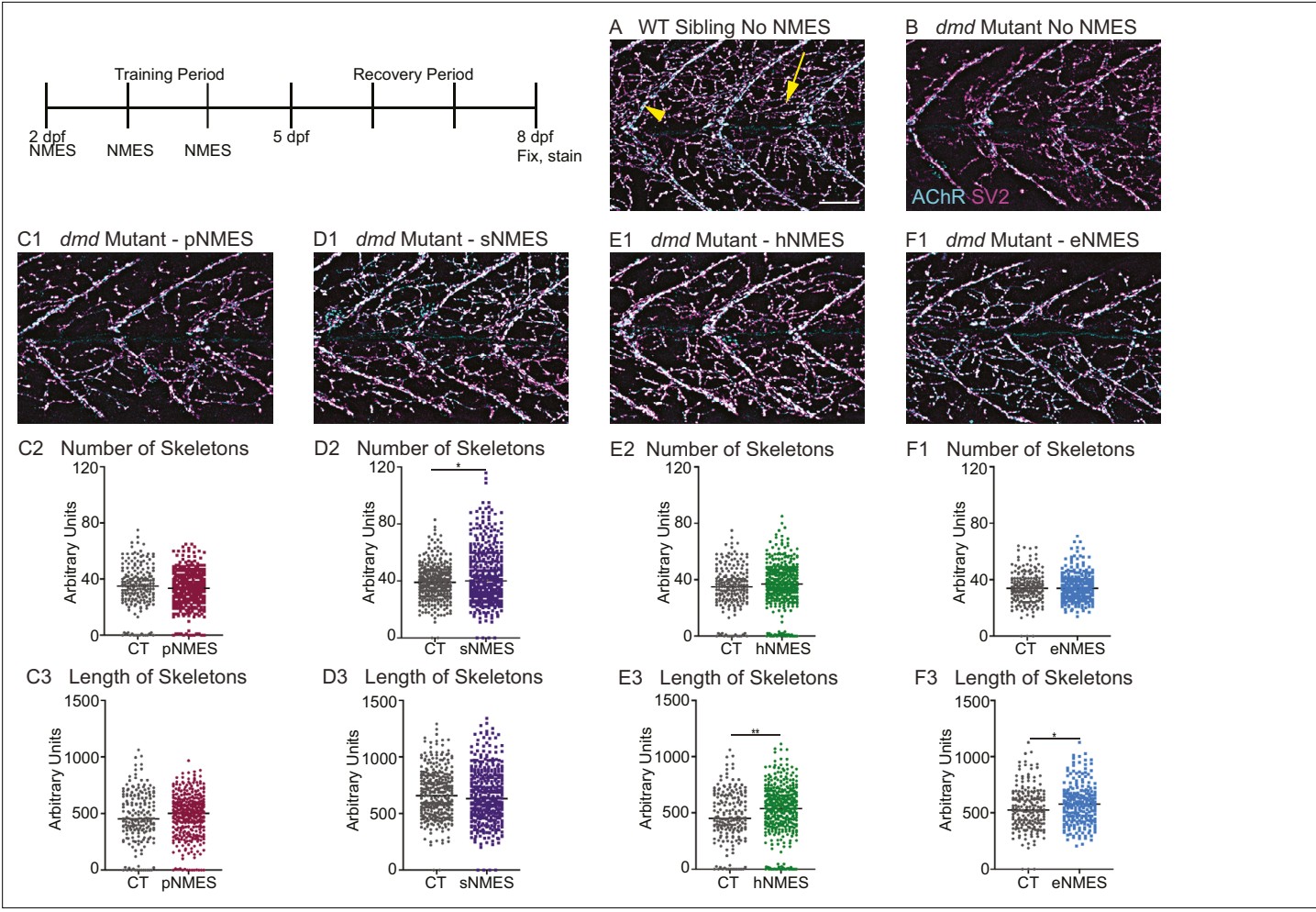

**Figure 4.** Impacts of neuromuscular electrical stimulation (NMES) paradigms on neuromuscular junction (NMJ) structure. Anti-SV2 (cyan) alpha-bungarotoxin (AChR; magenta) visualize the pre- and postsynaptic components of the NMJ, respectively. (**A**) Representative image of WT sibling. Myoseptal innervation, innervation at the chevron-shaped myotendinous junction (yellow arrowhead) is slow-twitch innervation. Fast-twitch muscle innervation is the network between the MTJs, yellow arrow points to fast-twitch muscle innervation. (**B**) Representative image of *dmd* mutant demonstrates a visible reduction in innervation, with relatively large portions of the muscle segments lacking innervation. (**C1, D1, E1, F6**) Representative images of *dmd* mutants that completed three sessions of the NMES paradigms. NMJ images were skeletonized as previously described. Strength NMES(sNMES) was the only paradigm that increased the number of skeletons (**D2**). Both hypertrophy NMES (hNMES) and endurance NMES (eNMES) increase skeleton length (**E3, F3**). Power NMES did not change the number or length of skeletons compared to *dmd* mutant controls (**C2,3**). Scale bar is 50 μm. NMJ data were analyzed using either an ordinary one-way ANOVA with Tukey's multiple comparisons test or a Kruskal–Wallis test with Dunn's multiple-comparison test. **p<0.01, ***p<0.001, ****p<0.0001.

alpha-bungarotoxin to stain postsynaptic AChR. We focused on analyzing fast-twitch muscle fiber innervation, which is called distributed innervation (the rich network of NMJs, *Figure 4A*, yellow arrow, in between the chevron-shaped slow-twitch muscle innervation at the myotendinous junctions [MTJs], *Figure 4A*, yellow arrowhead). Note that there are many spiderweb-like NMJs throughout the segments in WT larvae (*Figure 4A*). In contrast, there are fewer NMJs in *dmd* mutants (*Figure 4B*). Analysis was done in a semi-automated fashion that involved skeletonizing the NMJs as previously described (*Bailey et al., 2019*). pNMES did not change the number or length of skeletons (*Figure 4C*). sNMES increased the number of skeletons (*Figure 4D*) (n = 18 control, n = 20 sNMES; p=0.0395; one biological replicate). Both hNMES (*Figure 4E*) and eNMES (*Figure 4F*) did not increase the number of skeletons but increased skeleton length (n = 15 control, n = 23 hNMES, p=0.0036; n = 18 control, n = 15 eNMES, p=0.0278; one biological replicate). Thus, all NMES paradigms, other than pNMES, improved the number or length of NMJs compared to control *dmd* mutants.

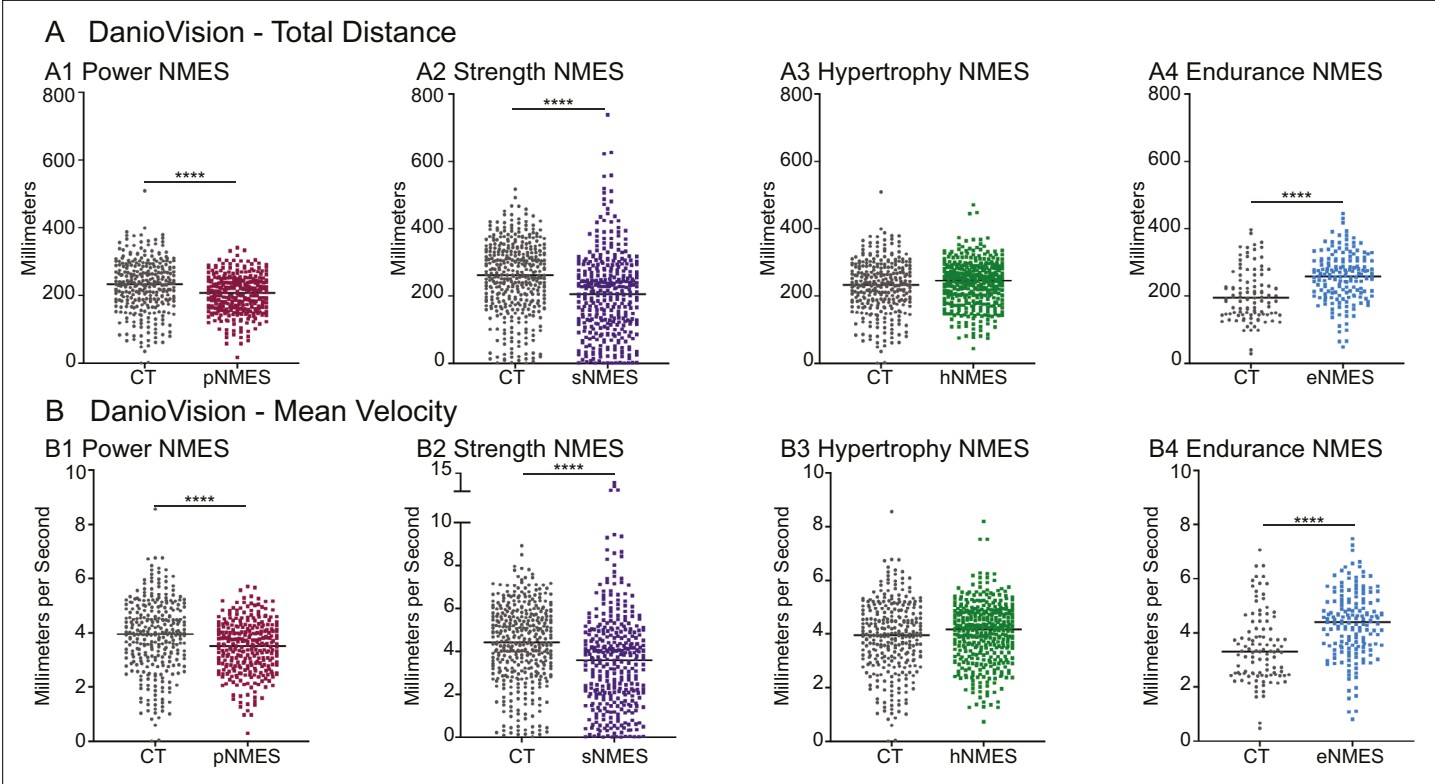

**Figure 5.** Impacts of neuromuscular electrical stimulation (NMES) paradigms on swimming. DanioVision was used to assess the impact of NMES on total distance (**A**) and (**B**) mean velocity. Measurements were made at 8 days post-fertilization (dpf). (**A1, B1**) *dmd* mutants that completed power NMES (pNMES) exhibited significant reductions in total distance and mean velocity compared to *dmd* mutants in the control group. (**A2, B2**) Strength NMES also negatively affected swimming activity in *dmd* mutants compared to control *dmd* mutants. (**A3, B3**) No change in total distance or mean velocity is observed following hypertrophy NMES (hNMES). (**A4, B4**) *dmd* mutants that completed endurance NMES (eNMES) swam a significantly greater total distance and at a significantly faster mean velocity compared to *dmd* mutants in the control group. Each data point represents a single time point for an individual zebrafish. Each zebrafish has a total of 15 points. DanioVision data were analyzed using two-sided *t*-tests. **p<0.01, ***p<0.001, ****p<0.0001.

The data presented thus far indicate that only eNMES improves both muscle and NMJ structure. We hypothesized that this improved neuromuscular structure would correlate with improved function. We tested this hypothesis by assessing swim activity as a gross readout of muscle function. Swim activity was tested using DanioVision (Noldus Information Technology). Swimming activity was recorded for 25 min with alternating 5 min light/dark periods. As predicted, eNMES was the only NMES paradigm that resulted in both increased distance (n = 7 control, n = 11 eNMES; p<0.0001; one biological replicate) and increased mean velocity (n = 7 control, n = 11 eNMES; p<0.0001; one biological replicate) compared to control *dmd* larvae (*Figure 5A4, B4*). pNMES negatively affected swimming activity (for both total distance and mean velocity: n = 20 control, n = 19 pNMES; p<0.0001; one biological replicate) (*Figure 5A1,B1*) despite having improved muscle structure. sNMES also significantly reduced total distance (n = 26 control, n = 23 sNMES; p<0.0001; one biological replicate) and mean velocity (n = 26 control, n = 23 sNMES; p<0.0001; one biological replicate) (*Figure 5A2, B2*) while hNMES did not affect these two measures (n = 20 control, n = 25 hNMES; p=0.1353 for total distance; p=0.1951 for mean velocity; one biological replicate) (*Figure 5A3, B3*).

Lastly, survival was tracked in *dmd* mutants treated with NMES. Survival checks were performed twice daily. Three sessions of eNMES (n = 32 control, n = 37 eNMES; p<0.0001; two biological replicates), sNMES (n = 63 control, n = 55 sNMES; p=0.0004; two biological replicates), and pNMES (n = 37 control, n = 32 pNMES; p=0.0414; one biological replicate) slightly but significantly extended the median age of survival for *dmd* mutants compared to unstimulated *dmd* mutants (*Figure 6A, B and D*); with eNMES having the largest beneficial effect. Three sessions of hNMES, however, did not affect median survival age (n = 37 control, n = 32 hNMES; p=0.6788; one biological replicate) (*Figure 6C*). Taken together, these data indicate that different NMES paradigms elicit different neuromuscular

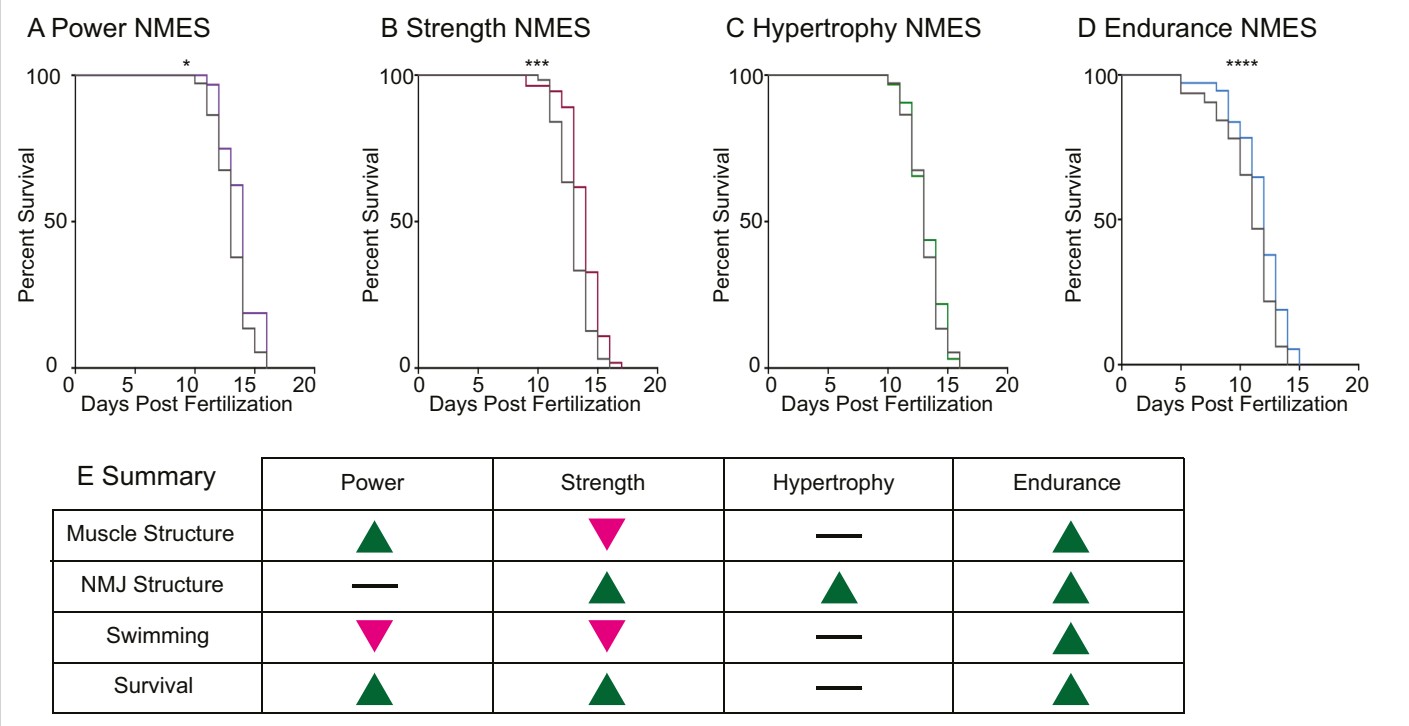

**Figure 6.** Impacts of neuromuscular electrical stimulation (NMES) paradigms on survival. Survival was tracked following completion of the three NMES sessions. Survival was significantly improved in *dmd* mutants that completed power (**A**), strength (**B**), and endurance (**C**) NMES. (**D**) Hypertrophy NMES had no effect on survival in *dmd* mutants. Survival data were analyzed using a Mantel–Cox test. *p<0.05, ***p<0.001, ****p<0.0001. (**E**) Summary of the impacts of NMES paradigms on neuromuscular structure and function. Note that endurance NMES is the only paradigm that improved all aspects.

responses. Out of the four NMES paradigms we tested, only eNMES improves neuromuscular structure, swimming, and life span (*Figure 6E*). Thus, we further investigated the impacts of eNMES on *dmd* muscle structure.

## eNMES improves muscle structure and sarcomere length

We found that eNMES significantly reduced muscle fiber detachments in *dmd* mutants (n = 22 control, n = 18 eNMES; p=0.0191; one biological replicate) (*Figure 3F2*). Many of the muscle segments in control *dmd* mutants had disorganized muscle fibers with a characteristic 'waviness,' and this appeared to be improved with eNMES. In order to quantify this aspect of muscle health, we used machine learning and trained the computer to identify, pixel-by-pixel, 'healthy' versus 'sick' with 97% accuracy. Next, we asked the computer to identify the percentage of healthy muscle in the same phalloidin images in which fiber detachments were counted on. This is visualized as muscle labeled either green (healthy) or red (sick) in *Figure 7A*. From this analysis, we observed that *dmd* mutants completing three sessions of eNMES trend towards having higher percentages of health muscle compared to control *dmd* mutants (n = 17 control, n = 15 eNMES; p=0.2052; one biological replicate) (*Figure 7A4*).

Sarcomere length impacts muscle function (*Moo and Herzog, 2018*). Sarcomeres produce force through the cross-bridges formed between actin and myosin, and the amount of force generated is dependent upon the amount of overlap between these thick and thin filaments (*Gordon et al., 1966*). We used second harmonic generation (SHG) microscopy to investigate sarcomere structure at 8 dpf. SHG is a nonlinear process in which two photons of frequency $\omega$ designated as (E * E) interact with the non-centrosymmetric aligned dipole ($\chi^{(2)}$) region of the myosin tail. A single photon with twice the frequency ($2\omega$) and half the wavelength is emitted in this energy-conserving label-free process. WT siblings exhibited a mean sarcomere length of 1.853 ± 0.1071 µm (*Figure 7B1, B4*). This result corresponded with sarcomere lengths previously published in 3 dpf WT zebrafish (1.86 ± 0.15 µm; *Huang et al., 2011*). Sarcomeres were significantly shorter in *dmd* larvae (n = 356 sarcomeres *dmd* control [four fish total], n = 282 sarcomeres WT sibling [three fish total]; p<0.0001; one biological replicate), with a mean length of 1.575 ± 0.1567 µm (*Figure 7B2, B4*). This result corresponded with previous

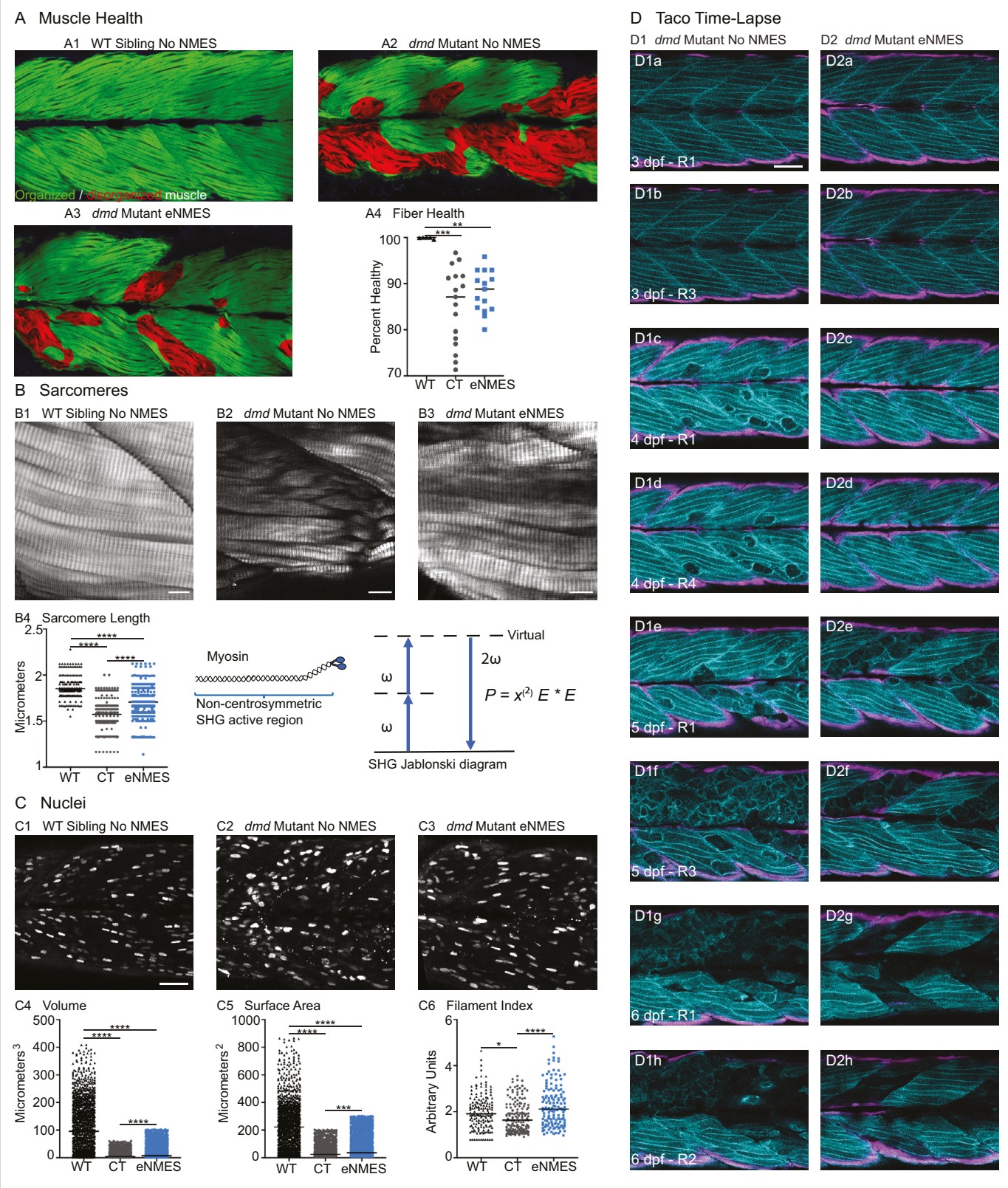

**Figure 7.** Endurance neuromuscular electrical stimulation (eNMES) improves multiple components of muscle health. (**A**) Machine learning was used to quantify muscle health pixel-by-pixel. Green indicates healthy pixels while red indicates unhealthy pixels. (**B**) Second harmonic generation (SHG) microscopy was used to quantify sarcomere length at 8 days post-fertilization (dpf). Representative SHG images of WT sibling control (**B1**), *dmd* mutant controls (**B2**), and *dmd* mutants that completed eNMES training. Anterior left, dorsal top, side mounted. Scale bars are 10 μm. (**B4**) Sarcomere

*Figure 7 continued on next page*

*Figure 7 continued*

length is significantly shorter in *dmd* mutant controls compared to WT sibling controls. However, eNMES significantly improves sarcomere length, bringing it closer to WT lengths. Each point represents a single sarcomere along a predetermined length of a muscle fiber. Multiple muscle fibers were measured per zebrafish. (**C**) Muscle nuclei were imaged at 8 dpf as a potential mechanism for improved muscle health. Anterior left, dorsal top, side mounted. (**C1**) Representative image of WT sibling control demonstrates healthy ellipsoidal nuclei organized along the length of the muscle fibers. (**C2**) Representative image of *dmd* mutant control demonstrates fragmented punctae as well as more spherical nuclei that clustering within the muscle segments. (**C3**) Representative image of *dmd* mutant that completed eNMES training demonstrates healthier, ellipsoidal nuclei that appear more organized within the muscle segments. Quantification of nuclear size indicates that eNMES significantly increases the volume (**C4**) and surface area (**C5**) of muscle nuclei compared to *dmd* mutant controls. However, nuclei are still significantly smaller compared to WT sibling controls, visually appearing to have an increased number of myonuclei compared to unstimulated *dmd* mutants. (**C6**) Filament index was used to assess circularity, specifically the departure from a circle. Filament index is significantly higher in *dmd* mutants that completed eNMES training, indicating that nuclei are more elongated compared to *dmd* mutant controls. Each point represents a single nuclei within a z-stack. (**D**) Transgenic *dmd* mutants (mylpfa:lyn-cyan [cyan], smych1:GFP [magenta]) were used to visualize changes in structural integrity of fast- and slow-twitch muscle fibers across three days. Anterior left, dorsal top, side mounted. Scale bar is 50 µm. Images were taken around the 12th myotome. (**D1**) Representative *dmd* mutant control. (**D1a–D1b**). At 3 dpf, there is no dystrophy in the imaged myotomes. (**D1c–D1e**) At 4 dpf and the beginning of 5 dpf, dystrophy is minimal with relatively few detaching muscle fibers. (**D1f**) However, massive muscle degeneration occurs between the first found of imaging and the third round of imaging at 5 dpf. (**D1g–D1h**) Fiber degeneration is still present, suggesting that the damaged muscle fibers have not been cleared and regeneration is unlikely. (**D2**) Representative *dmd* mutant that is undergoing eNMES training. (**D2a–D2b**) The first session of eNMES at 3 dpf does not result in immediate damage to the muscle. (**D2c–D2d**) Similarly, following the second session of eNMES at 4 dpf, there is no immediate muscle damage occurring in the imaged myotomes. (**D2e**) At 5 dpf, following the third session of eNMES, muscle fiber degeneration is evident, but by the third round of imaging (**D2f**), these damaged areas are being cleared and there is evidence of regeneration. (**D2g–D2h**) At 6 dpf, previously damaged muscle segments have new muscle fibers present. All data were analyzed using either an ordinary one-way ANOVA with Tukey's multiple comparisons test or a Kruskal–Wallis test with Dunn's multiple-comparison test. *$p<0.05$, **$p<0.01$, ***$p<0.001$, ****$p<0.0001$.

studies that showed shorter sarcomeres in *dmd* larvae (*Widrick et al., 2016*). We tested the hypothesis that eNMES would increase sarcomere length. Three sessions of eNMES significantly increased mean sarcomere lengths ($1.707 \pm 0.1710$ µm) compared to *dmd* mutant controls (n = 356 sarcomeres *dmd* control [four fish total], n = 550 sarcomeres eNMES [four fish total]; $p<0.0001$; one biological replicate) (*Figure 7B3, B4*). Although eNMES-treated *dmd* larvae still had shorter sarcomeres than WT siblings, these data suggest the hypothesis that eNMES may improve muscle structure and function by restoring sarcomere lengths to more optimal lengths.

## Muscle nuclei return to a more ellipsoidal shape with eNMES

Myonuclear size and shape play an important role in muscle health (*Folker and Baylies, 2013*; *Roman and Gomes, 2018*). We measured three components of muscle nuclei size and shape: volume, surface area, and filament index. Filament index is a measure that quantifies the departure of an object from a circle. A circle has a filament index of 1, and a higher filament index indicates a departure to a more ellipsoidal shape. Thus, a higher filament index indicates that nuclei are more elongated, which is suggested to be healthier (*Bruusgaard et al., 2003*). Muscle nuclei in *dmd* mutants had significantly lower volumes (n = 1417 nuclei *dmd* control, n = 1355 nuclei WT sibling; $p<0.0001$; data shown for one fish), surface areas (n = 1451 nuclei *dmd* control, n = 1378 nuclei WT sibling; $p<0.0001$; data shown for one fish), and filament indices (n = 156 nuclei *dmd* control, n = 158 nuclei WT sibling; $p=0.0187$; data shown for one fish) compared to WT siblings (*Figure 7C4–6*). Interestingly, eNMES significantly increased these measures (volume: n = 1417 nuclei *dmd* control, n = 1861 nuclei eNMES, $p<0.0001$; surface area: n = 1451 nuclei *dmd* control, n = 1990 nuclei eNMES, $p<0.0001$; data shown for one fish), especially for filament index (n = 156 nuclei *dmd* control, n = 140 nuclei eNMES; $p<0.0001$; data shown for one fish), which is restored to WT values (*Figure 7C6*). Additionally, these nuclei appear more organized along the length of individual muscle fibers (*Figure 7C3*), similar to the pattern observed in WT siblings. These data suggest that *dmd* mutants have smaller, spheroidal nuclei compared to WT siblings, and eNMES is capable of elongating the nuclei, and increasing their volumes and surface areas.

## Longitudinal confocal analysis suggests less degeneration and more hypertrophy/regeneration with eNMES

We used transgenic zebrafish (a generous gift from Drs. Sharon Amacher and Jared Talbot; *Hromowyk et al., 2020*) to visualize muscle structure through time. Disease onset in these transgenic zebrafish is at

3 dpf; thus, NMES sessions were conducted at 3, 4, and 5 dpf while the recovery period extended from 6 through 9 dpf. At 3 dpf, there was not a clear difference in muscle degeneration between treated and control mutants (*Figure 7D1a, b, D2a, b*). However, by 4 dpf, control mutants exhibited initial signs of muscle degeneration (*Figure 7D1c, D1d*). eNMES mutants showed less degeneration, suggesting that eNMES delays degeneration (*Figure 7D2c, D2d*). Whereas degenerated fibers persist in control mutants for days (*Figure 7D1f,g*), degenerated segments are cleared more quickly in eNMES-treated mutants (*Figure 7D2f,g*). Finally, more robust hypertrophy or regeneration was observed in eNMES-treated mutants (*Figure 7D2h*). Taken together, these data suggest that eNMES improves muscle homeostasis.

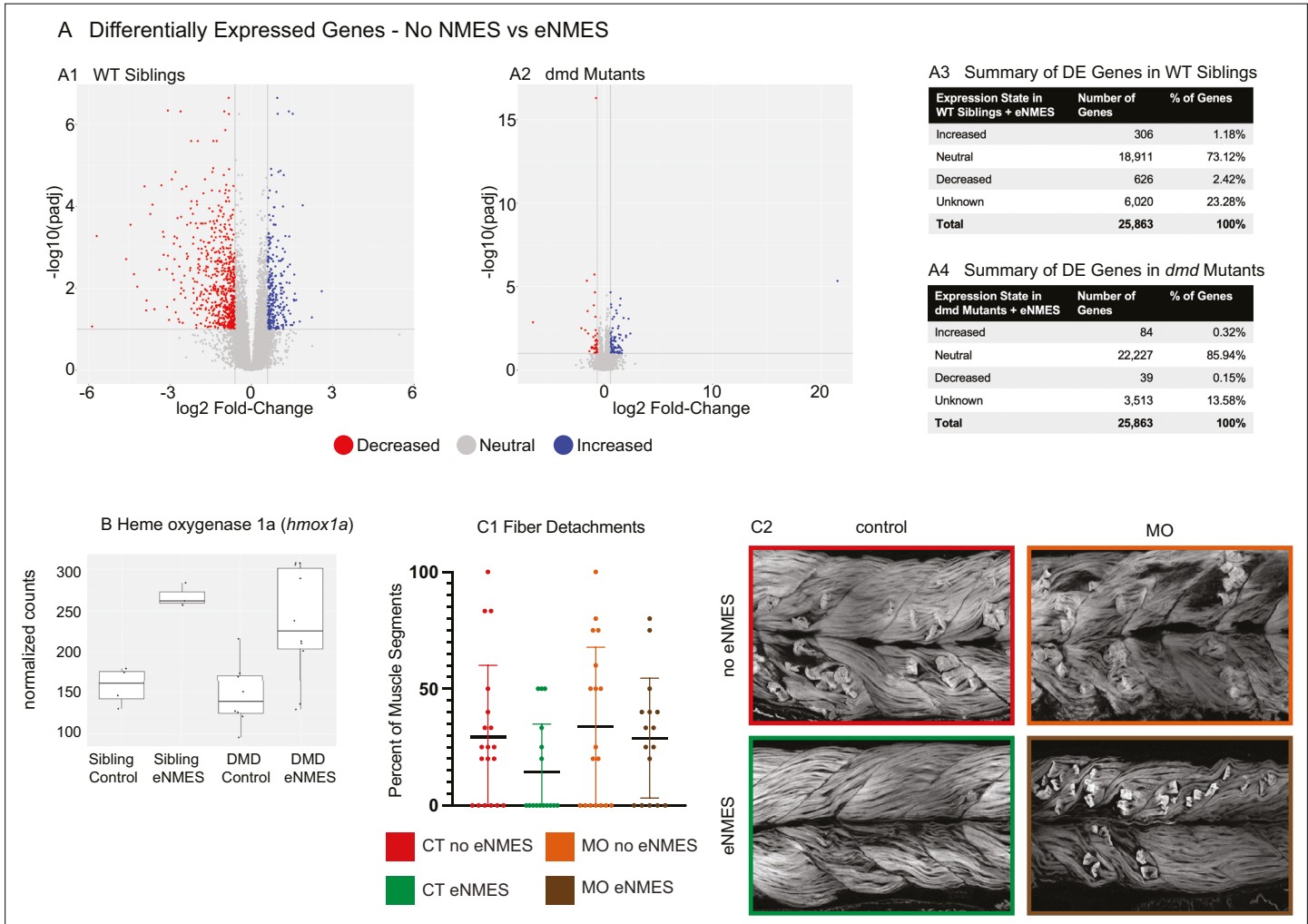

**Figure 8.** *dmd* mutants do not respond to endurance neuromuscular electrical stimulation (eNMES) in the same manner as WT siblings and heme oxygenase is required for eNMES-mediated improvement. RNAseq analysis was performed at 7 days post-fertilization (dpf) in WT siblings and *dmd* mutants that completed eNMES training, and their expression patterns were compared with their respective controls. (**A1**) Volcano plot showing significantly and biologically upregulated (blue dots) and downregulated (red dots) genes in WT siblings that completed eNMES versus those that did not. (**A2**) Volcano plot showing significantly and biologically upregulated (blue dots) and downregulated (red dogs) genes in *dmd* mutants that completed eNMES versus those that did not. (**A3–A4**) Summary of differentially expressed genes in WT siblings (**A3**) and *dmd* mutants (**A4**). WT siblings had 932 differentially expressed genes compared to 123 differentially expressed genes in *dmd* mutants, suggesting that *dmd* muscle responds differently to eNMES and the genes responsible for eliciting beneficial effects on muscle structure and function are different. (**B**) *hmox1a* expression was increased with eNMES in both WT and *dmd* mutants. (**C1**) Whereas eNMES significantly reduces the percentage of segments with dystrophy, *dmd* mutants injected with morpholinos against *hmox1a* do not show improvement with eNMES. (**C2**) Representative images of larvae showing that *hmox1a* is necessary for eNMES-mediated improvement of *dmd* muscle.

## *dmd* mutants may respond differently than WT controls to eNMES

RNAseq was conducted on RNA extracted at 7 dpf, 3 days following the last NMES session. The broad conclusion from this experiment is that eNMES does not appear to act primarily through transcriptional regulation, at least when assessed 3 days after the last NMES session. Principal component analysis (PCA) revealed that WT siblings that underwent three sessions of eNMES cluster separately from those in the control group (data not shown), suggesting that WT siblings that complete three sessions of eNMES have a unique expression profile compared to WT controls. Of the total 25,863 genes identified across the RNAseq analysis, 932 genes were differentially expressed between WT siblings in the eNMES versus control groups (*Figure 8A1*). Of these 932 genes, 306 genes were increased and 626 genes were decreased (*Figure 8A3*). Twenty-four Gene Ontology (GO) terms were identified, including regulation of metabolic processes, regulation of MAP kinase activity, regulation of transcription, and circadian rhythms.

Interestingly, the admittedly small number of differentially expressed genes suggests that eNMES may be eliciting changes in WT versus *dmd* mutants through different mechanisms. Based on the number of genes differentially expressed, *dmd* mutants do not respond to NMES in the same manner as WT siblings (*Figure 8A2*). 123 genes were differentially expressed (false discovery rate [FDR] < 0.1 and abs(log2 (fold change)) > 0.6) between *dmd* mutants that completed three eNMES sessions versus *dmd* mutant controls (*Figure 8A4*). This number is much lower than the 932 differentially expressed genes in WT siblings. Additionally, *dmd* mutants have more genes that were increased (n = 84) than decreased (n = 39), which is the opposite of WT siblings, further suggesting that *dmd* mutants do not respond through the same signaling pathways as their healthy counterparts. Unfortunately, GO analyses did not reveal specific cellular processes in which these genes may participate in to positively impact muscle health. Only 4 of the 1048 differentially expressed genes elicited by eNMES in both *dmd* mutants and WT siblings shared the same expression pattern (not shown). This lack of overlap between differentially expressed genes further indicates that *dmd* mutants do not respond similarly to eNMES as WT siblings.

## Heme oxygenase is necessary for eNMES-mediated improvement

Heme oxygenase (HO) is an antioxidant that has been implicated as a potential therapeutic treatment in both zebrafish and mouse models of *dmd* (*Chan et al., 2016*; *Kawahara et al., 2014*). HO 1a was upregulated by eNMES in both WT and *dmd* larvae (*Figure 8B*). We asked whether HO was necessary for the eNMES-mediated improvement in *dmd* mutants by using previously published morpholinos (*Kawahara et al., 2014*). Morpholinos were injected at the one-cell stage, and *dmd* mutants injected with *hmox1a* morpholinos were treated with eNMES. Not surprisingly, control *dmd* mutants treated with eNMES had fewer fiber detachments (*Figure 8C*) and more organized muscle (*Figure 8D*). In contrast, *dmd* mutants injected with morpholinos against *hmox1a* showed no improvement after eNMES treatment (*Figure 8C and D*). Birefringence, percentage of segments with degenerating fibers, and mobility were all unchanged with eNMES in *dmd* mutants injected with *hmox1a* morpholinos (*Figure 8B and C*, data not shown). These data suggest that HO is necessary for eNMES-mediated improvement. Interestingly, however, when we analyzed RNAseq data of transcripts known to be involved in *hmox1a* regulation, we found that eNMES did not induce changes in these transcripts (data not shown). This result further suggests that the primary impact of eNMES is not through transcriptional regulation.

## eNMES reduces susceptibility to contraction-induced disruption of muscle structure in *dmd* mutants

The ECM surrounding muscle fibers is a critical component of muscle fiber health. Protein complexes spanning the sarcolemma and ECM serve as mechanical linkages and signaling hubs that promote muscle plasticity (*Csapo et al., 2020*). However, excess ECM protein deposition can also lead to fibrosis. We asked whether ECM proteins are differentially expressed in zebrafish *dmd* larvae following eNMES. Despite the fact that RNAseq data represent a snapshot in time and are not the best way to capture a structure as dynamic as the ECM, we observed changes in ECM gene expression with eNMES that suggest that eNMES could impact ECM deposition. Transforming growth factor beta induced (TGFBI) is an ECM protein that binds to type I, II, and IV collagens as well as several integrins. *Tgfbi* is upregulated in *mdx* muscle compared to healthy muscle (*Coles et al.,*

*2020*; *Pescatori et al., 2007*). We found that *tgfbi* is also significantly higher in zebrafish *dmd* mutants compared to WT controls. Expression of *tgfbi* in both *dmd* mutants and WT siblings was reduced with eNMES (*Figure 9A1*). Periostin (postnb) is a TGFBI-related protein that is involved in modeling the ECM and connective tissue architecture during development and regeneration, serving specifically as a mediator of fibrosis in injury and disease (*Ozyilmaz et al., 2019*). RNAseq data indicate that *postnb* shares a similar expression pattern with *tgfbi*: increased expression in *dmd* mutants compared to WT siblings and a reduction in this expression following eNMES in both groups (*Figure 9A2*). These data suggest the hypothesis that fibrosis could be reduced in eNMES-treated zebrafish. One impact of a reduction in excess fibrosis could be increased muscle-cell adhesion to the ECM.

Cell-matrix adhesion is negatively affected in various models of muscular dystrophy, and restoration of adhesion improves muscle structure and function (*Burkin et al., 2005*; *Burkin et al., 2001*; *Goody et al., 2012*). Therefore, the downregulation of key cell adhesion proteins following eNMES was puzzling and led us to ask whether muscle cell-matrix adhesion was altered by eNMES. We did this by subjecting zebrafish to a hard stimulation paradigm designed to make muscle fibers detach from their ECM for two back-to-back sessions (*Figure 9B*). This experiment was conducted 2 days after the final eNMES training session. This 1 min hard stimulation paradigm was defined by a frequency of 4 pulses per second, a delay of 60 ms, a duration of 2 ms, and a voltage of 30 V, which is similar to that known to initiate muscle fiber detachment from their ECM (*Subramanian and Schilling, 2014*). Birefringence images were taken before and after each session. To ensure consistency in imaging, zebrafish were mounted laterally with their left side facing up, and the same imaging parameters were used for each zebrafish across all imaging sessions. We then analyzed the change in mean gray values before stimulation compared to after the first or second session. Nearly half of the control mutants (10/22) had decreased mean gray values after the first session (*Figure 9C4*), and slightly over half (13/22) had decreased mean gray values after the second session (*Figure 9C5*). In contrast, just under 25% of eNMES-treated mutants (5/22) had a decreased mean gray value after the first session (*Figure 9C4*) and slightly under a third had a decreased mean gray value after the second session (7/22; *Figure 9C5*). While there are no differences in absolute mean gray values between control and eNMES mutants before and after the first round of stimulation (n = 22 control, n = 22 eNMES; p=0.3453; one biological replicate) (*Figure 9C4*), the change in mean gray values for eNMES-treated *dmd* mutants trends higher (healthier muscle) than controls following the second round of stimulation (n = 22 control, n = 22 eNMES; p=0.0803; one biological replicate) (*Figure 9C5*). There are slightly fewer muscle segments with detachments in *dmd* mutants that completed eNMES compared to control *dmd* mutants (n = 22 control, n = 18 eNMES; p=0.2505; one biological replicate) (*Figure 9D4*). The most striking difference in appearance between the control and eNMES *dmd* mutants after the hard stimulation was the improved organization of muscle fibers in eNMES-treated *dmd* mutants. Whereas control *dmd* mutants had lots of disorganized fibers (*Figure 9D2a*, red arrow), eNMES *dmd* mutants had more organized fibers (*Figure 9D3a*, green arrow). We used machine learning to quantify overall muscle health. This approach showed that *dmd* mutants that completed eNMES had a significantly higher percentage of healthy muscle compared to control *dmd* mutants (n = 21 control, n = 22 eNMES; p=0.0496; one biological replicate) (*Figure 9D5*). Taken together, these data indicate that eNMES-treated *dmd* mutants can withstand contraction-induced disruption of muscle structure better than *dmd* mutant controls.

## Integrin alpha7 is required for eNMES-mediated improvement in muscle structure

The above data suggest the hypothesis that muscle fiber adhesion to the matrix is increased in *dmd* mutants treated with eNMES. Itga7 is a transmembrane receptor that mediates the response of skeletal muscle to eccentric exercise (*Boppart et al., 2008*; *Lueders et al., 2011*; *Mahmassani et al., 2017*). It is not known whether Itga7 mediates the response to NMES. We hypothesized that Itga7 is required for eNMES-mediated improvement. We tested this hypothesis by generating Itga7 mutants (*Coffey et al., 2021*) and testing whether eNMES improves muscle structure in these mutants. We found that eNMES did not impact muscle structure in *itga7-/-* larvae (*Figure 9E*). Thus, Itga7 is required for eNMES-mediated improvement of muscle structure at least in the context of *itga7* mutants.

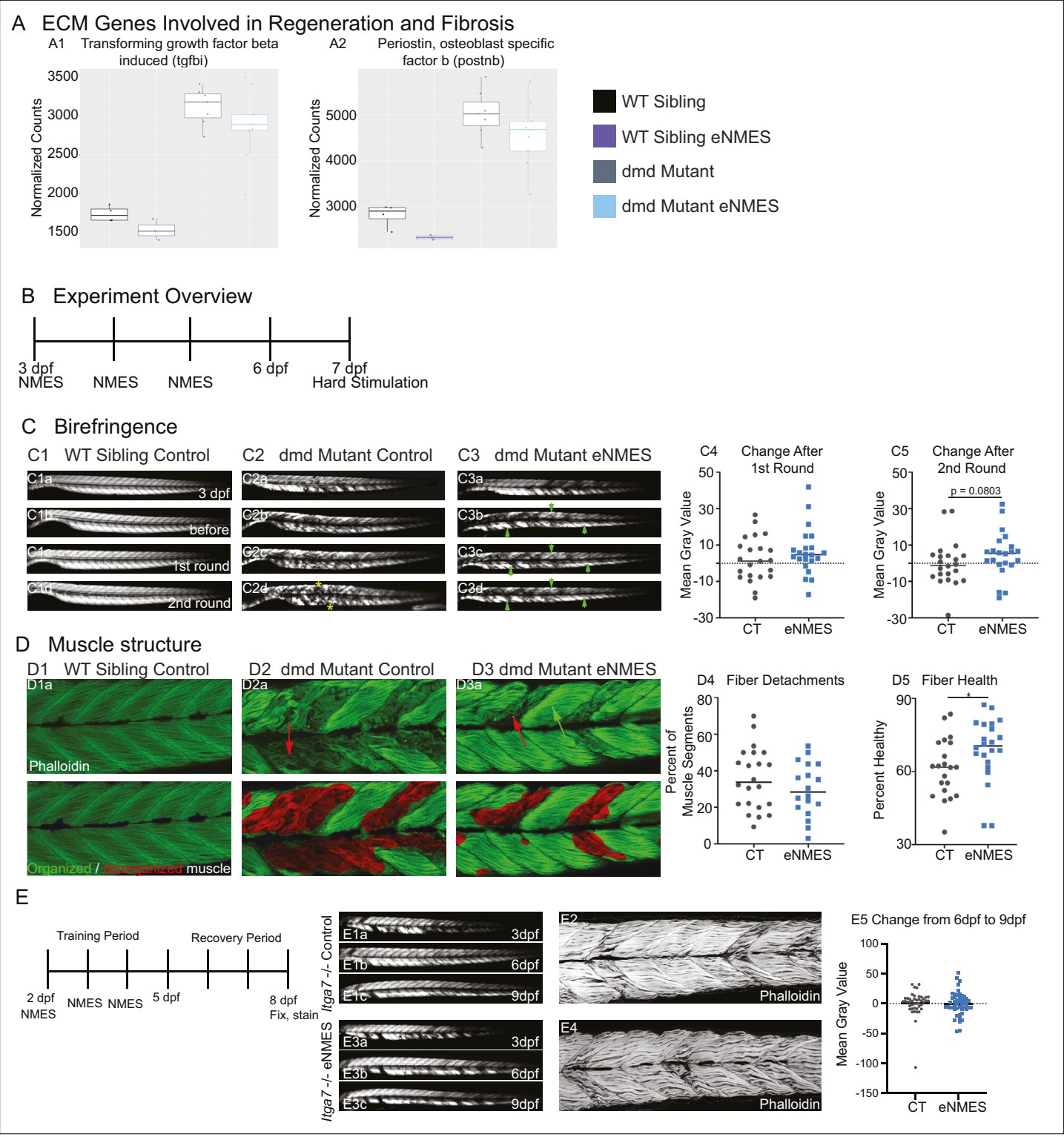

**Figure 9.** Muscle resilience to hard stimulation is increased with endurance neuromuscular electrical stimulation (eNMES), and Itga7 is required for eNMES-mediated improvement. We identified three extracellular matrix (ECM) genes from RNAseq analysis, *tgfbi* (**A1**), *postnb* (**A2**), *itgb1b.2* (not shown) that are significantly upregulated in *dmd* mutants compared to WT siblings and trend towards being downregulated with eNMES in *dmd* mutants. (**B**) Experimental overview. At 3 days post-fertilization (dpf) (disease onset), birefringence images were taken followed by the first session of eNMES. At 4 and 5 dpf, zebrafish undergo the second and third NMES sessions, respectively. At 7 dpf, muscle resilience was tested using a hard electrical stimulation paradigm intended to cause muscle damage. (**C**) Birefringence images were taken at 3 dpf (**C1a**). (**C1b–d**) Birefringence images

*Figure 9 continued on next page*

*Figure 9 continued*

were taken at 7 dpf before the first hard stimulation (**C1b**), after the first hard stimulation (**C1c**), and after the second hard stimulation (**C1d**). No visible changes in birefringence are observed in WT siblings after the two stimulation sessions. (**C2**) For *dmd* mutant controls, the first round of stimulation did not result in visible changes to birefringence (**C2c**), but, after the second round, areas of muscle degeneration are visible (**C2d**, yellow asterisks). Conversely, in *dmd* mutants that completed three sessions of eNMES, the first (**C3c**) and second (**C3d**) rounds of stimulation did not result in visible changes to birefringence (green arrowheads denote intact areas of birefringence that remain intact). (**C4, C5**) Change in birefringence from before to after the first round (**C4**) and second (**C5**) of stimulation suggests that eNMES training may improve muscle resilience. (**D**) Phalloidin was used to visualize individual muscle fibers. (**D1a**) Representative image of a WT sibling control demonstrates healthy, organized muscle fibers, and myotomes. (**D2a**) Representative image of a *dmd* mutant control highlights disorganized and wavy muscle fibers and fiber detachments. (**D3a**) Representative image of a *dmd* mutant that completed eNMES demonstrates some wavy muscle fibers and detached fibers intermixed with relatively healthy myotomes. (**D4**) The percent of muscle segments with detached fibers following the hard stimulation is reduced in *dmd* mutants that complete eNMES training compared to *dmd* mutant controls. For this analysis, a muscle segment was defined as half of a myotome. (**D1b, D2b, D3b**) Machine learning was used to quantify muscle health pixel-by-pixel. Green indicates healthy pixels while red indicates unhealthy pixels. (**D5**) The percent of healthy muscle following the hard stimulation is significantly higher in *dmd* mutants that completed eNMES compared to *dmd* mutant controls. (**E**) *itga7* mutants were subjected to the same eNMES protocol that results in improvements in *dmd* mutants. Note that eNMES does not improve birefringence (panels **E1, E3**, quantified in **E5**) or muscle structure in *itga7* mutants (**E2, E4**). All data were analyzed using two-sided *t*-tests. *p<0.05.

## Discussion

We used an experimental design that leverages the power of the zebrafish model's ability to perform in vivo analyses of numerous components of organismal health across time in individual zebrafish. By implementing this longitudinal design, we demonstrate that (1) different NMES paradigms elicit different effects on neuromuscular structure, swimming, and life span; (2) eNMES positively benefits neuromuscular health, function, and survival in *dmd* mutants; (3) changes are accompanied by improvements in NMJ length, nuclear shape and size, and sarcomere lengths; (4) *dmd* mutants respond to NMES differently than WT siblings; (5) HO signaling is required for eNMES-mediated improvement; and (6) Itga7 is required for eNMES-mediated improvement, suggesting that cell adhesion is increased in eNMES-treated embryos (*Figure 10*). These findings indicate that the zebrafish model is a valuable tool for studying skeletal muscle plasticity and that healthy and dystrophin-deficient muscle use different mechanisms to maintain homeostasis.

### The impacts of activity on the progression of DMD

Reviews regarding the potential impact of exercise on DMD muscle tend to draw the same conclusion: that more research with studies that incorporate longitudinal designs, different modes of exercise, impacts of exercise on other treatment modalities, and standardized outcome measures is necessary (*Anziska and Sternberg, 2013*; *Hyzewicz et al., 2015*; *Markert et al., 2012*; *Markert et al., 2011*; *Voet et al., 2013*). Treadmill exercise is frequently used in mouse models to exacerbate the *mdx* phenotype (*Hyzewicz et al., 2015*). However, there are multiple studies that show beneficial effects of either treadmill exercise or voluntary wheel running on the progression of muscle degeneration in *mdx* mice (*Gaiad et al., 2017*; *Zelikovich et al., 2019*). It is important to note that the vast majority of these studies investigated aerobic activity, with only one 'resistance training' regimen that involved adding weights to a running wheel. Thus, the impact of resistance training is not well understood. With regards to aerobic exercise, the true answer is likely that there is both some variation among individuals and that there is a delicate balance between positive and negative impacts of exercise on muscle homeostasis. Our data showing that different NMES regimes have different impacts on muscle structure and function support the hypothesis that there is not a clear 'one-size-fits-all' approach to exercise and *dmd*.

### Zebrafish as a model for elucidating neuromuscular plasticity

The negative consequences of inactivity on muscle resilience led us to ask whether the activity could improve muscle resilience, and, therefore, disease progression. We selected NMES as a mechanism to elicit consistent, repeatable contraction patterns across individual zebrafish. We generated four NMES paradigms that varied in frequency and voltage to test how different contraction patterns impact muscle structure, function, and survival. Collectively, our experiments suggest that *dmd* muscle exhibits a delicate, intricate equilibrium with several factors influencing muscle structure, swimming activity, and survival. Birefringence does not predict swimming performance and swimming

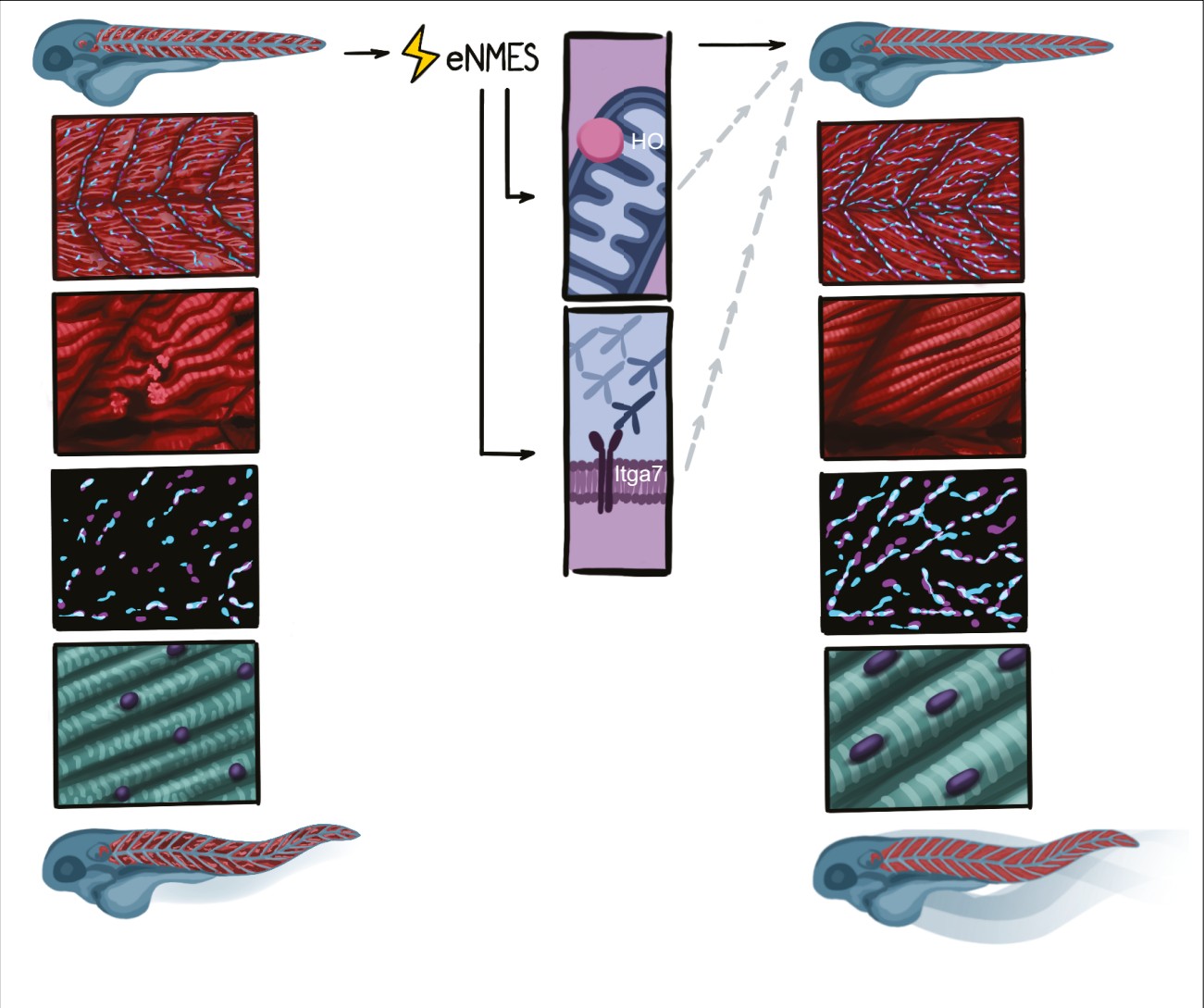

**Figure 10.** Summary. Endurance neuromuscular electrical stimulation (eNMES) positively benefits neuromuscular health, function, and survival in *dmd* mutants. Muscle fibers in *dmd* mutants treated with eNMES are more organized and have fewer detachments. Neuromuscular junctions (NMJs) are longer in eNMES-treated *dmd* mutants. Sarcomeres are longer and nuclei are more ellipsoid and aligned. eNMES-treated mutants swim faster and more distance than control siblings. Both heme oxygenase and Itga7 are required for eNMES-mediated improvements.

performance does not predict survival. For example, two paradigms, eNMES and pNMES, improved muscle structure while two paradigms, hNMES and sNMES, negatively affected muscle structure. Surprisingly, though, only eNMES increased swimming activity. In contrast, survival was extended by eNMES as well as pNMES and sNMES. Therefore, this is a new model to understand disease progression and elucidate mechanistic pathways that target improvements in structure, function, and survival.

## Potential benefits of NMES

Electrical stimulation has been shown to be generally safe and potentially effective for some conditions. For example, there are potential therapeutic benefits of NMES for the treatment of spinal cord injuries. Although not all trials observe an increase in voluntary muscle strength with NMES, none found deleterious effects of NMES (*de Freitas et al., 2018*). NMES can also improve dysphagia after stroke: 10 out of 11 trials showed that NMES improved swallowing with only one showing no effect (*Alamer et al., 2020*). NMES combats disuse atrophy in multiple contexts. Chronic NMES applied to mice who were anesthetized for 2.5 weeks not only showed increased muscle mass in the stimulated limb, but also had improved insulin sensitivity (*Lotri-Koffi et al., 2019*). NMES for at least 7 days is

sufficient to improve muscle mass of lower limbs in non-ambulatory patients with traumatic brain injury (*Silva et al., 2019*). NMES is even being studied as a means to combat muscle atrophy during spaceflight (*Maffiuletti et al., 2019*). The above data show that NMES has potential benefits for sudden muscle disuse caused by external events, but the molecular and cellular mechanisms are not well understood.

NMES also shows promise for neurodegenerative disorders and aging muscle. Muscle mass and strength are improved in aged rats with NMES, and NMES improves muscle mass and balance in older adults as well as older adults with dementia (*Dow et al., 2005*; *Kern et al., 2014*; *Nishikawa et al., 2021*). NMES improves mobility in patients with progressive multiple sclerosis (*Wahls et al., 2010*). NMES may also improve mobility and strength in Amyotrophic Lateral Sclerosis (ALS) (*Handa et al., 1995*), although the intensity may be important (*Group, 2017*). Far less is known about NMES in the context of muscular dystrophies. The concept of super-imposing electrical stimulation to improve dystrophic muscle was proposed by the neurologist who first described DMD over a hundred years ago (*Duchenne, 1870*). Despite the longevity of this hypothesis, it has not been sufficiently tested as a therapy for DMD. There are promising data: NMES improves muscle fiber morphology in dystrophic mice (*Dangain and Vrbova, 1989*; *Luthert et al., 1980*; *Vrbová and Ward, 1981*) and chickens (*Barnard et al., 1986*). In chickens, the benefit was most pronounced if administered prior to rampant muscle degeneration (*Barnard et al., 1986*). Small trials in young children also suggest that early low-frequency NMES can improve voluntary muscle contraction compared with the contralateral leg (*Scott et al., 1990*; *Scott et al., 1986*). NMES can also improve muscle function in myotonic dystrophy (*Chisari et al., 2013*) and limb girdle muscular dystrophy (*Kilınç et al., 2015*). Despite these promising studies, NMES is not commonly used as an adjuvant therapy in myopathies and dystrophies. This is potentially due to the impractical approach of chronic NMES for most if not all skeletal muscles. Thus, it is important to elucidate the underlying molecular and cellular mechanisms of beneficial impacts of NMES. It is known that electrical stimulation increases both the number and size of AChR clusters in primary myoblasts. The fact that the expression of Rhapsyn is also increased indicates that the increased AChR clusters are leading to increased mature NMJs. However, clearly more mechanistic studies regarding the effects of NMES on muscular dystrophies are warranted.

## Potential mechanisms for improved neuromuscular function

We found that eNMES increased sarcomere lengths, which could improve force generation by leading to a more optimal interaction between actin and myosin filaments. Nuclear volume, surface area, and filament index were increased with eNMES, suggesting that muscle nuclei are returning to their elongated shape. As nuclear size affects DNA organization, transcriptional and translational processes, and nuclear import and export activities (*Levy and Heald, 2012*), minor changes in size correlate with reduced muscle function and fiber performance (*Windner et al., 2019*). Therefore, these improvements in muscle nuclei may also mediate improvements in muscle structure and function following eNMES. Time-lapse imaging data support the hypothesis that eNMES is creating an environment that supports regeneration. Following eNMES in *dmd* mutants, there is less degeneration and in those areas with degenerating fibers, newly regenerated fibers appear sooner. RNAseq data largely suggested that the impacts of eNMES are not primarily transcriptional. The RNAseq data did however identify two potential mechanisms that may allow for the above improvements to occur. HO is an antioxidant that has been implicated as a potential therapeutic treatment in both zebrafish and mouse models of *dmd* (*Chan et al., 2016*; *Kawahara et al., 2014*). Hmox1a expression was upregulated in both WT and *dmd* mutants with eNMES. We found that *hmox1a* is necessary for the eNMES-mediated improvement in muscle structure and function in *dmd* mutants.

The second mechanism includes the potential remodeling of the ECM. The ECM is constantly responding to signals from both within and outside the cell, and incorporating these signals to create a scaffold that supports either regeneration or fibrosis such that the cell is protected from further damage. Our RNAseq data suggest that eNMES may result in ECM remodeling to support regeneration and/or limit fibrosis. One impact of changes in ECM could be increased adhesion of muscle fibers to their ECM. Fiber cross-sectional area is increased in transgenic mice overexpressing Itga7 after eccentric exercise (*Lueders et al., 2011*; *Zou et al., 2011*). In contrast, muscle damage is exacerbated in Itga7 mutant mice, especially near sites of high mechanical force near the MTJs (*Boppart et al., 2008*). While the complete mechanisms are not known, Itga7 promotes enhanced proteostasis

(*Mahmassani et al., 2017*) and increased Sca-1$^+$CD45$^-$ mesenchymal stem cells. We asked whether Itga7 was required for eNMES-mediated improvement. Itga7 mutant larvae did not improve with eNMES, indicating that Itga7 is required for eNMES-mediated improvement. We did not generate *dmd;itga7* double mutants to test whether eNMES is not effective in this context, which would be interesting to do. Taken together, these data suggest the hypothesis that one mechanism of eNMES improvement is increased adhesion of muscle fibers to their ECM.

## Summary

Identifying the basic mechanisms by which activity impacts muscle health in the context of muscle disease is a crucial first step towards identifying potential therapies. Here, we identify an NMES paradigm that improves neuromuscular structure, function, and life span. We show that NMES differently affects gene expression in WT versus *dmd* mutants. This result indicates that it is critical to study the impacts of the activity on diseased muscle in addition to WT muscle. Finally, we show that eNMES acts via Itga7 and HO signaling. Taken together, our data not only establish a model system for neuromuscular plasticity in healthy versus diseased muscle but also clearly elucidate the beneficial effects of NMES.

## Materials and methods

### Zebrafish husbandry and transgenic lines

Zebrafish embryos were retrieved from natural spawns of adult zebrafish maintained on a 14 hr light/10 hr dark cycle. We used sapje$^{ta222a}$ zebrafish (*Bassett et al., 2003*). For live imaging studies, we used transgenic sapje$^{ta222a}$ 3MuscleGlow zebrafish expressing mylpfa:lyn-cyan, smych1:GFP, and myog:H2B:RFP (gift from Drs. Sharon Amacher and Jared Talbot; *Hromowyk et al., 2020*). Embryos were grown in embryo-rearing media (ERM) with methylene blue at 28.5°C. Embryos were manually dechorionated at 1 dpf. Zebrafish were fed once daily beginning at 5 dpf (Larval AP100 Dry Larval Diet [<50 μm], Zeigler, PA). For survival studies, zebrafish were housed in 20 mm Petri dishes with 10 mL of system water per dish beginning at 8 dpf. Survival checks were performed in the morning and at night. All protocols conform to the University of Maine Institutional Animal Care and Use Committee's Guidelines.

### Experimental overview

Experiments were conducted identically so that variables such as treatment duration, disease stage at the time of treatment, and disease stage at the time of evaluation did not change. Zebrafish were followed individually throughout each experiment so that disease progression could be monitored throughout time in longitudinal studies. Experiments began at disease onset. Disease onset for our *sapje* line is at 2 dpf. For the live imaging studies, disease onset in the transgenic *sapje* line is at 3 dpf even though the alleles harboring the mutation are identical. However, this transgenic line was imported from The Ohio State University, while our line has been maintained solely at the University of Maine. Experiments were carried out exactly the same for our *sapje* line and the transgenic line. At disease onset, zebrafish were identified via birefringence as a *dmd* mutant or healthy WT sibling. Healthy WT siblings had myotomes with organized, parallel muscle fibers that appear bright white while *dmd* mutants had myotomes with disorganized and detached muscle fibers that appear gray to black (*Bassett et al., 2003*). Larvae were housed in 12-well plates (one fish per well) with 3 mL of ERM per well. Zebrafish were then randomly assigned to control or NMES cohorts for the next 3 days (the treatment period). At the end of this treatment period, zebrafish were allowed to recover for an additional 4 days (recovery period). During the treatment and recovery periods, disease progression was monitored by daily birefringence and swim activity analyses, with a special emphasis on what is occurring at 5 and 8 dpf as these mark the beginning and end of the recovery period.

### NMES paradigm

The first NMES session began at disease onset. The treatment period included one session of NMES at 2, 3, and 4 dpf for a total of three sessions. Following the completion of the third NMES session, zebrafish entered the recovery period from 5 to 8 dpf.

Zebrafish were subjected to NMES in groups of four using our 3D printed 'gym' (*Figure 1B*). The rectangular gym is divided into six rectangular wells that measure 4.7625 mm (length), 1.5875 mm (width), and 1.5875 mm (depth). Two tunnels run parallel to the smaller sides of the rectangular wells and the positive and negative electrodes slide through these tunnels such that they are exposed only in the wells. This allows the delivery of electrical pulses to each zebrafish simultaneously. Prior to the NMES session, zebrafish were transferred to tricaine solution (612 µM in 1× ERM) for 4 min. At the end of the 4 min, zebrafish were placed into a well with its head facing the positive electrode and its tail facing the negative electrode. The positive and negative electrodes were attached to a Grass SD9 Stimulator, which was used to generate the electrical pulses. The frequency, delay, and voltage were adjusted for the different paradigms (*Figure 1C and D*). Each NMES session lasted 1 min. Following each NMES session, zebrafish were removed from the gym and placed back into their respective wells.

## Birefringence analysis

Birefringence was used to quantitatively assess the daily progression of dystrophy (*Berger et al., 2012*). Zebrafish were placed in tricaine (612 µM) immediately prior to imaging and then transferred to a 35 mm glass-bottom dish. Birefringence images were taken on a Leica MZ10 F Stereomicroscope with a Zeiss AxioCamMRm or Leica DMC5400 camera attached. An analyzer in a rotatable mount (Leica) was attached to the objective, and the glass-bottom Petri dish was placed on the polarized glass stage. Images were taken at the same time every day within an experiment. Imaging parameters were consistent for all zebrafish and across all days. Mean gray values were calculated using Fiji software as described previously (*Berger et al., 2012*). Briefly, the body of the zebrafish was outlined from the 6th to the 25th myotome using the 'Polygon selections' tool, and then the mean gray value was measured. Three separate outlines were drawn to obtain three separate measures, and the average was used for calculations. All images were blinded prior to measurements using a Perl script. Mean gray values are presented as a percentage of the average mean gray value of healthy WT siblings in the control group.

## DanioVision analysis

The DanioVision system and EthoVision XT 13.0 software (Noldus, VA) were used to conduct high-throughput locomotion tracking studies to better characterize the impact of NMES on zebrafish swim function. A clear 12-well plate was placed into the DanioVision observation chamber. The temperature control unit was set to 28.5°C, ensuring that the temperature of the ERM in the well plate was maintained. Zebrafish had a 5 min acclimation period to the observation chamber prior to the beginning of the recording period. Using the EthoVision software, we created a white-light routine that included 5 min in the dark followed by two light-on/off cycles, where the white light was turned on at 100% intensity for 5 min and then turned off for 5 min. The total recording time was 25 min. Recordings were made at the same time each day. For each fish, the average total distance and mean velocity across 1 min intervals were calculated, such that each fish had a total of 25 measurements for total distance and mean velocity. We then focused on swim activity during the three 5 min dark periods, which represent when zebrafish are most active. This analysis provided 15 measurements for each zebrafish.

## Membrane permeability indicated by Evan's blue dye

Evan's blue dye is a membrane-impermeable dye used to assess membrane damage. In *dmd* mutants, EBD is used to assess muscle fiber integrity, and we used EBD to assess fiber integrity pre- and post-NMES using the methods described by *Smith et al., 2015*. EBD (Sigma-Aldrich, MO) was dissolved to 1% w/v in 0.9% saline solution. This EBD stock solution was further diluted to 0.1% then loaded into an injection needle pulled from glass capillary tubes on a Sutter Flaming/Brown Micropipette Puller (CA). Zebrafish were placed in tricaine (612 µM) for 4 min. At the end of the 4 min, zebrafish were aligned on a 1% agarose-lined Petri dish in a minimal volume of ERM. The needle was gently inserted into the pericardial space, and EBD was injected using a MPPI-3 pressure injector (ASI, Eugene, OR). Zebrafish were allowed to recover for 4 hr, providing ample time for the dye to circulate the body and enter damaged muscle fibers. Zebrafish were prepared for live imaging as described above for birefringence. An ET DSR fluorescent filter (Leica) was used to visualize EBD. After imaging the initial dye amount in each zebrafish, zebrafish underwent one session of NMES as described above. Immediately after the NMES session, zebrafish were again prepared for live imaging. This allowed us to observe

whether NMES caused additional dye entry into the muscle. Imaging parameters remained the same for all zebrafish and imaging sessions. Zebrafish were mounted laterally with the head on the left and dorsal up. To quantify EBD entry, we calculated mean gray values using the same methods described for birefringence except the outline was drawn from the first visible somite to the last visible somite. All images were blinded prior to analysis using a Perl script. Data is presented as the average mean gray value for three separate measurements.

## Immunostaining

Zebrafish were fixed in 4% paraformaldehyde for 4 hr at room temperature. After fixation, embryos were rinsed in PBS-0.1% Tween 20 (PBS-tw; Bio-Rad, Hercules, CA). For visualizing muscle structure, phalloidin was used. Zebrafish were first permeabilized in PBS-2% Triton-X-100 (Fisher Scientific, Waltham, MA) for 1.5 hr and then placed in 1:20 phalloidin (Invitrogen, Eugene OR) in PBS-tw for 4 hr on the rocker at room temperature. Zebrafish were rinsed out of phalloidin using PBS-tw and stored in PBS-tw until imaged. For visualizing NMJs, zebrafish were stained with alpha-bungarotoxin and SV2. Zebrafish were first permeabilized in 1 mg/ml collagenase in 1× PBS for 1.5 hr, and then stained with 1:500 alpha-bungarotoxin-647 (Invitrogen) and 1:20 phalloidin in antibody block (5% BSA [Fisher Scientific], 1% DMSO [Sigma-Aldrich], 1% Triton-X-100, 0.2% saponin from quillaja bark [Sigma-Aldrich] in 1× PBS) for 2 hr at room temperature. Zebrafish were rinsed using PBS-tw and placed in antibody block overnight at 4°C. Zebrafish were then stained with 1:50 SV2 (DSHB, Iowa City, IA) in antibody block for 3 days at 4°C. Upon removal from SV2, zebrafish were rinsed using PBS-tw and then placed in antibody block for 8 hr on the rocker at room temperature. This was followed by an overnight incubation in 1:200 GAM (Invitrogen) in antibody block. Zebrafish were then rinsed out of secondary antibody using PBS-tw and stored in PBS-tw until imaged. Phalloidin-488 or -546 and GAM-488 or -546 were used interchangeably with no differences in staining observed.

## Imaging

Confocal imaging was used to visualize phalloidin and NMJ staining. Fixed and stained zebrafish were deyolked and then mounted in a 24-well glass-bottom plate using 0.5% low-melt agarose (Boston BioProducts, Ashland, MA) in 1× PBS. For live confocal time-lapse imaging, zebrafish were anesthetized in tricaine solution (612 µM in 1× ERM) for 4 min and then mounted in a 24-well glass-bottom plate or 30 mm glass-bottom Petri dish using 0.5% low-melt agarose in 1× ERM (with 612 µM tricaine). Two or three zebrafish were placed in each well. Zebrafish were mounted anterior left and dorsal up to ensure the same side of the fish was imaged each day. Finally, a small amount of tricaine solution (612 µM in 1× ERM) was added to prevent the agarose from evaporating and ensure the zebrafish remained anesthetized throughout the imaging session. Upon completion of imaging, zebrafish were gently removed from the agarose using a fine fishing line and returned to their respective wells. All fluorescent images were captured using the Leica SP8 confocal microscope.

SHG imaging was used as a label-free mechanism to visualize sarcomeres. Fixed zebrafish were deyolked and then mounted in a 30 mm glass-bottom Petri dish using 1.0% low-melt agarose in 1× PBS. Once the agarose solidified, the Petri dish was filled with 1× PBS. Images were acquired using a custom-built two-photon microscope. This system uses a modified Olympus FV300 system with an upright BX50WI microscope stand and a mode-locked Ti:Sapphire (Coherent Ultra II) laser. Laser power was modulated via an electro-optic modulator. The SHG signals were collected in a non-descanned geometry using a single PMT. Emission wavelengths were separated from excitation wavelengths using a 665 nm dichroic beam splitter followed by a 448/20 nm bandpass filter for SHG signals. Images were acquired using circular polarization with excitation power ranging from 1 to 50 mW and a 40 × 0.8 NA water immersion objective with 3× optical zoom and scanning speeds of 2.71 s per frame. All images were 512 × 512 pixels with a field of view of 85 µm.

## Image analysis

All images were blinded using a Perl script prior to analysis. The percent of myotomes with muscle fiber detachments was calculated manually by counting the number of muscle segments with visibly detached fiber(s). Muscle segments are defined as half myotomes. Additionally, we used machine learning to identify healthy versus unhealthy muscle fibers. For these analyses, we used MATLAB to implement a deep learning approach to segment images of phalloidin-stained fish into healthy

muscle, sick muscle, and background. We used the DeepLab v3+ system with an underlying Resnet18 network (*Chen et al., 2017*). We defined the ground truth dataset manually using LabelBox (https://labelbox.com/). Training images and ground truth images were broken down into 256 × 256 pixel images for training. The training dataset was divided into 60% training, 20% validation, and 20% test data. Median frequency weighting was used to balance the classes. Each fish was oriented such that the head of the fish would be at the left of the image. Data was augmented to translate the images by 10 pixels vertically and horizontally. Rotation was found to make the network less accurate as the orientation angle of the muscle fibers relative to the body orientation is important to assess their health. The stochastic gradient descent with momentum optimizer was selected with 0.9 momentum. The maximum number of epochs was 100, and the mini-batch size was 8. In every epoch, the training dataset was shuffled. The number of iterations between evaluations of validation metrics was 315. The patience of validation stopping of network training is set up to 4. The initial learning rate used for training was 0.001. The learning rate was dropped 0.3-fold piecewise during training every 10 epochs. The factor for L2 regularization (weight decay) was 0.005. The training set reached an accuracy of 97%. Images were then segmented by the MATLAB *semanticseg* command, which produced eight-bit unsigned integer segmentations. The fraction of each fish that was determined to be healthy was reported as a fraction of the total muscle. Pixels determined to be background (i.e., not muscle) were excluded from this calculation.

For NMJ analyses, we used a method previously published by our laboratory (*Bailey et al., 2019*). To prepare images for analysis, a custom Fiji macro was written to keep image processing consistent throughout all experiments. First, the raw .lif file was opened in Fiji and the image was split into its respective channels (phalloidin, AChR, and SV2). The phalloidin channel was immediately saved as a .tif file and closed. For the AChR and SV2 channels, duplicate z-stacks were created and a 10-pixel radius Gaussian blur was applied. These blurred images were then subtracted from their original images, respectively. The resulting images were then merged to a single image and a maximum intensity projection was generated. This maximum intensity projection was saved as a .tif file and closed. For each experiment, the maximum intensity projections were combined into a single .tif file using a custom MATLAB script. This combined .tif file was then opened in Fiji and three separate masks, marking the fish body, horizontal myoseptum, and myoseptal innervation, were drawn on the projected images using the Pencil tool. These masks were used to define individual muscle segments, where each muscle segment represents half of a single myotome. Using a custom MATLAB script, skeleton number and skeleton length were calculated for each muscle segment across all zebrafish analyzed.

Muscle nuclei were analyzed using Fiji's 3D Objects Counter as well as the Measure tool. To prepare images for analysis, we first reduced background noise by duplicating the z-stack, performing a 10-pixel Gaussian blur on the duplicated image, and subtracting the blurred image from the original image. We then performed a 1-pixel Gaussian blur on the resultant image and set a threshold using 'max entropy' setting. With this image, we used the Analyze Particles tool to generate masks to use with the 3D Objects Counter tool as well as the Measure tool. The 3D Objects Counter tool provided surface area and volume measurements while the Measure tool provided perimeter, area, and major axis measurements, which were used to calculate filament index.

To calculate sarcomere lengths, SHG images were first imported into ImageJ, and then, using the Freehand selection tool, two lines were drawn to indicate the outer boundaries (top and bottom) of the muscle fiber being analyzed. The Freehand selections were converted into .txt files and imported into LabVIEW VI. Using LabVIEW, the midline of the two selections (top and bottom) was determined. The midline was then imported back into ImageJ over the original photo such that it was positioned in the center of the sarcomeres. Next, the Plot Profile tool and Peak Finder tool were used to determine the peaks, which correspond to sarcomere length. Since the Peak Finder tool gives the distance in pixels, a conversion factor was used to convert pixels to micrometers based on the objective and optical zoom used. Multiple muscle fibers are analyzed for each zebrafish. We avoided the optical illusion effect of ESH veneers regions when measuring sarcomere lengths (*Dempsey et al., 2015*).

## RNA extraction and RNAseq

Total RNA was extracted from whole zebrafish at 7 dpf from replicate samples using the Zymo Direct-zol RNA microprep kit. Each biological replicate consisted of two zebrafish. For WT siblings,

there were 4 replicates for the control group and 3 replicates for the eNMES group, and for *dmd* mutants, there were 8 replicates for the control group and 10 replicates for the eNMES group. Prior to performing RNA extractions, zebrafish within the eNMES and control groups were grouped based on their severity at disease onset and the calculated change in their birefringence from 5 dpf to 7 dpf. RNA was kept at –80°C until it was shipped to Quick Biology (Pasadena, CA) for sequencing. Following RNA quality control using an Agilent BioAnalyzer 2100 (), polyA+RNAseq libraries were prepared for each sample using the KAPA Stranded RNA-Seq Kit (KAPA Biosystems, Wilmington, MA). Final library quality and quantity were analyzed by Agilent Bioanalyzer 2100 and Life Technologies Qubit3.0 Fluorometer. Each library was sequenced using 150 bp paired-end reads using an Illumina HiSeq4000 (Illunnia Inc, San Diego, CA).

Analyses of RNAseq reads were completed on the Advanced Computing Group Linux cluster at the University of Maine. To determine the quality of the RNA sequencing reads before further processing, FastQC version 0.11.7 was utilized. Following this quality assessment, reads were concatenated tail-to-head to produce one forward FASTQ file and one reverse FASTQ file for each sample. These FASTQ files were then trimmed of adapter sequences, and low-quality leading and trailing ends were removed using Trimmomatic version 0.36.0 (*Bolger et al., 2014*). Trimmed paired-end reads mapped to the Ensembl-annotated zebrafish transcriptome (Ensembl version 95) to generate read counts per gene using RSEM version 1.2.31 (*Li and Dewey, 2011*) with bowtie version 1.1.2 (*Langmead et al., 2009*). Read counts were analyzed using the DESeq2 version 1.22.2 (*Love et al., 2014*) to analyze gene expression, p-value, and FDR. Genes with fewer than te10 mapped reads across all samples were excluded. For each pairwise comparison of treatment groups, differentially expressed genes were determined using FDR p-value cutoff of 0.1 and requiring at least a 0.6 $\log_2$ fold change (in either direction). Resulting lists were used for GO enrichment analysis and set analysis for each pairwise comparison.

Sets of differentially expressed genes (both increased and decreased expression) were analyzed to test for enriched GO Biological Process terms (FDR < 0.1) using GOrilla (http://cbl-gorilla.cs.technion.ac.il/). For this analysis, the entire set of expressed genes were used as a background. In cases where GOrilla found no enriched terms, PantherDB's overrepresentation test on Biological Processes (http://pantherdb.org/) was used. Again, the entire set of expressed genes list was used as the background, and results were evaluated using Panther's Fisher's exact test and p-values were adjusted for multiple testing using FDR.

Ensembl gene IDs were mapped to gene symbols and names using zebrafishMine'sAnalyse feature (http://www.zebrafishmine.org/). In some cases, manual mapping was used by comparing Zfin.org gene search and Ensembl gene search results. Summarized gene expression data are available at the Gene Expression Omnibus (accession number GSE155465), and FASTQ files are available at the Short Read Archive (accession number SRP274405).

## Cell adhesion

Muscle fiber attachment strength was assessed similarly to that published by *Subramanian and Schilling, 2014*. Zebrafish larvae were anesthetized with tricaine (612 µM in 1× ERM) for 4 min and then placed in the NMES gym. The stimulator settings were adjusted such that the frequency was four pulses per second, the delay was 60 ms, the duration was 2 ms, and the voltage was 30 V. Zebrafish were stimulated for 1 min. Birefringence images were taken pre- and post-stimulation as was described for the EBD study. Zebrafish were then subjected to a second round of stimulation, and birefringence images were taken after this second round (*Figure 9*). Zebrafish were mounted laterally with the head on the left and dorsal up and the same imaging parameters were used for all zebrafish.

## Statistical analysis

All statistical analyses were performed in GraphPad Prism. Normality was first assessed for all data using the Shapiro–Wilk test. If data passed this normality test, an unpaired two-tailed *t*-test was performed between two datasets (i.e., *dmd* mutant control versus *dmd* mutant eNMES) while an ordinary one-way ANOVA was performed followed by a Tukey's multiple comparison test between three datasets (i.e., WT sibling control versus *dmd* mutant control versus *dmd* mutant eNMES). Conversely, if data failed the Shapiro–Wilk normality test, a Mann–Whitney *U* test was performed for comparing

two datasets while a Kruskal–Wallis test was performed for comparing three datasets. Significance for all tests was set to p<0.05.

## Acknowledgements

We thank Drs. Sharon Amacher and Jared Talbot for developing the transgenic 3MuscleGlow zebrafish and sharing this valuable tool with us; Dr. Joy-El Talbot at Iris Data Solutions for her expertise in RNAseq analysis; NVIDIA Corporation for donating the Quadro P6000 used for deep learning analyses; Keegan Kilroy for designing the NMES gym and assistance with designing the NMES paradigms; and Mark Nilan for exceptional zebrafish care at the UMaine Zebrafish Facility.

## Additional information

### Funding

| Funder | Grant reference number | Author |
| --- | --- | --- |
| Morgan Hoffman Foundation | | Elisabeth A Kilroy |
| National Science Foundation | GRFP | Elisabeth A Kilroy |
| University of Maine | Start up | Karissa Tilbury |
| University of Maine | Seed Grant | Benjamin L King Clarissa Henry |
| National Institutes of Health | R15 GM128026 | Joshua B Kelley |
| National Science Foundation | MRI 1726541 | Clarissa Henry |
| National Institutes of Health | RO1 AR075836 | Clarissa Henry |
| National Institutes of Health | R15 HD99018-01 | Clarissa Henry |

The funders had no role in study design, data collection and interpretation, or the decision to submit the work for publication.

### Author contributions

Elisabeth A Kilroy, Conceptualization, Data curation, Formal analysis, Funding acquisition, Investigation, Methodology, Project administration, Supervision, Writing – original draft, Writing – review and editing; Amanda C Ignacz, Formal analysis, Investigation, Visualization, Writing – review and editing; Kaylee L Brann, Claire E Schaffer, Devon Varney, Jordan N Miner, Karissa Tilbury, Formal analysis, Investigation; Sarah S Alrowaished, Tashawna L Spellen, Alexandra D Lewis, Investigation; Kodey J Silknitter, Formal analysis; Ahmed Almaghasilah, Formal analysis, Software; Benjamin L King, Data curation; Joshua B Kelley, Methodology, Software; Clarissa A Henry, Conceptualization, Data curation, Formal analysis, Funding acquisition, Project administration, Supervision, Visualization, Writing – original draft, Writing – review and editing

### Author ORCIDs

Jordan N Miner (ID) http://orcid.org/0000-0003-2998-3330
Clarissa A Henry (ID) http://orcid.org/0000-0001-7204-9231

### Ethics

This study was performed in strict accordance with the recommendations in the Guide for the Care and Use of Laboratory Animals of the National Institutes of Health. All of the animals were handled according to approved institutional animal care and use committee (IACUC) protocol A2020-06-01 of the University of Maine.

Decision letter and Author response
Decision letter https://doi.org/10.7554/eLife.62760.sa1
Author response https://doi.org/10.7554/eLife.62760.sa2

## Additional files

### Supplementary files
• Transparent reporting form

• Source data 1. Source data are organized by figure with titles of the measurements in the columns and sheets are named with the appropriate figure. Titles of panels also include the figure panel that the data is for.

### Data availability
Summarized gene expression data are available at the Gene Expression Omnibus (accession number GSE155465), and FASTQ files are available at the Short Read Archive (accession number SRP274405).

The following datasets were generated:

| Author(s) | Year | Dataset title | Dataset URL | Database and Identifier |
|---|---|---|---|---|
| King BL | 2020 | Expression profiling by high throughput sequencing | https://www.ncbi.nlm.nih.gov/geo/query/acc.cgi?acc=GSE155465 | NCBI Gene Expression Omnibus, GSE155465 |
| King BL | 2020 | FASTQ files | https://www.ncbi.nlm.nih.gov/sra/?term=SRP274405 | NCBI Sequence Read Archive, SRP274405 |

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
