## [Editor Report]

This is an interesting and well-conceived study that explores the potential benefit of electrical stimulation for muscular dystrophy in terms of muscle structure and motor function. The authors take advantage of the zebrafish model system, and a well-characterized zebrafish mutant that models Duchenne muscular dystrophy, to show that certain stimulation paradigms can improve muscle morphology and muscle performance, like via integrin-mediated pathway(s). The potential implications of this research are broad as they begin to address the key question in the MD field about whether and what types of exercise may (or may not) be beneficial to dystrophic muscle.

---

## [Decision Letter]

**Decision letter after peer review:**

Thank you for submitting your article "Inactivity is deleterious and neuromuscular stimulation is beneficial in an animal model of Duchenne Muscular Dystrophy" for consideration by *eLife*. Your article has been reviewed by 3 peer reviewers, and the evaluation has been overseen by a Reviewing Editor and Didier Stainier as the Senior Editor. The reviewers have opted to remain anonymous.

The reviewers have discussed the reviews with one another and the Reviewing Editor has drafted this decision to help you prepare a revised submission.

Summary:

The authors seek to tackle the question of exercise and inactivity in Duchenne muscular dystrophy, an important and unsolved issue. They use the zebrafish model system and two paradigms, one an inactivity paradigm (using tricaine) and the other an exercise paradigm using NMES. They find that inactivity worsens the dystrophic phenotype, and that different exercise paradigms impact the dystrophic phenotype differently. Overall this is an important study with exciting data and a potential to impact our understanding of exercise in DMD. However, as described below, all reviewers felt that several critical experimental considerations are necessary to consider in order to substantiate the data claims.

Essential Revisions:

1. The inactivity paradigm (e.g. figure 2) using tricaine as a means of inducing inactivity has pluses and minuses. There are issues with comparing it to rodent and human inactivity experiments (which usually involve hindlimb/limb immobilization), as the authors here are using chemical inhibition. Tricaine has systemic effects on multiple tissue types and organ systems including neurological and respiratory systems. I would be careful to call this model an inactivity model. Other appropriate models of inactivity exist, including physically restraining the zebrafish larvae to prevent movement and use of chemicals like BTS. In sum, the authors need to rule out if the consequences of tricaine administration is due to inactivity or pulmonary/secondary dystrophic pathology issues (e.g. swim bladder or respiration), and also consider a second "inactivity" paradigm in order to validate that the findings are due to inactivity.

2. NMJ changes are hypothesized as an explanation for the response to NMES paradigms. Can/have the authors evaluated the functional output of the NMJ in the NMES-treated DMD zebrafish? Were any electrophysiological measurements performed on the NMES treated DMD fish, independent of any therapeutic experimental protocol?

3. Hmox1 overexpression has been pursued as a strategy for DMD in mice by the Zoltan Arany and Joseph Dulak's groups, so the findings in figure 10 are supported. Have the authors evaluated whether or not the entire Hmox1 pathway was affected in the NMES-treated DMD fish?

4. For data presented in figure 1: authors describe the birefringence phenotype in mild mutants as increased degeneration for three days and then increased regeneration. Could they provide any experimental evidence of "muscle regeneration" mentioned in this statement?. Similarly, they mention severe dmd mutant regenerated throughout this study, however, no experimental data is provided to support this statement. As myotome contains both normal and degenerating myofibers, could improvement in birefringence be a consequence of the growth of those normal myofibers vs regeneration of sick myofibers? The term regeneration has also been used later in NEMS studies and needs to be supplemented with the experimental evidence of regeneration. In sum, there needs to be experimental support for the supposition of regeneration throughout the manuscript, as well as more careful consideration of rates of degeneration and regeneration.

5. There were several concerns with the transcriptomic data. More information related to the comparison of WT vs untreated sap is necessary for the interpretation of the changes seen with NMES. Also, the authors see very few consistent changes in terms of genes regulated one way in untreated sap and the other with treatment. It appears in fact that an overarching conclusion is that the transcriptional changes are NOT a driving force of the response to NMES. There is concern about the interpretability of taking one or two changed genes, as most transcriptional programs do not function in this isolated manner. For example, what is the true meaning of a change in b1 integrin without any changes in a integrin levels. In other words, how is this finding really of biological significance? In addition, the authors propose ECM changes as a potential mechanism. This should be supported by evaluation of ECM proteins (by western or immunostaining for example).

6. The authors clearly demonstrate that the phenotype is variable. However, they do not carry this variability forward to all of their inactivity and exercise studies. Several of those studies feature relatively low n numbers. How have the authors ensured that the differences seen are not, in fact, due to the natural variability of the dmd phenotype in zebrafish? In particular, this seems like it would be an issue for all studies with n sizes less than 20, particularly given that the magnitude of difference for many of the studies is small.

7. DMD is caused by damage in sarcolemma and subsequent myofiber detachment. The authors didn't observe any effect on myofiber structure but still found reduced velocity in mutants that were subjected to intermittent inactivity. Could this be due to a slight increase in sarcolemma damage (not examined here) and/or changes in the calcium in muscle fibers? Similarly, what are the effects of extended inactivity on MTJ structure? While authors make good observations with their animal model (as also seen in human and other animal models previously), mechanistic details underlying these changes are lacking.

[Editors’ note: further revisions were suggested prior to acceptance, as described below.]

Thank you for resubmitting your work entitled "Beneficial impacts of neuromuscular electrical stimulation on muscle structure and function in the zebrafish model of Duchenne Muscular Dystrophy" for further consideration by *eLife*. Your revised article has been evaluated by Didier Stainier (Senior Editor) and a Reviewing Editor.

The manuscript has been improved but there are some remaining issues that need to be addressed, as outlined below:

There are three major areas of concern related to the resubmitted manuscript.

(1) The authors fail to truly contextualize their findings with the abundant literature related to neuromuscular electrical stimulation in neuromuscular disease. This needs to be incorporated into the Discussion section in order to best interpret the current study within the field. Please see the comments from reviewer 1 below.

(2) The authors suggest that NMES may be working through modification of ECM-cell adhesion. They present data showing failure of itga7 mutants to respond to NMES. While this provides evidence that itga7 is potentially involved in mediated response in WT fish, it does not inform on the situation with dmd. The ideal experiment would of course be to test NMES in the setting of itag7/dmd double knockouts. At the very least, the fact that this link is not firmly established in the dmd model needs to be pointed out.

(3) The authors also implicate TGFb signalling based on RNAseq data. The authors rightly point out the short comings of RNAseq for uncovering the important pathways; this is particularly true for TGFb signalling, which is very much regulated and governed at the level of post translational changes. There are simple assays for examining TGFb signalling and activity that have been utilized in other DMD models (such as mdx) and in zebrafish. Such assays would lend support to the RNAseq data, which on its own its relatively weak proof of the involvement of the ECM and TGFb.

Additional specific reviewer comments can be found below.

*Reviewer #1:*

I appreciate the authors for taking several of my considerations and concerns in the newly incorporated revised manuscript. In particular the BTS study showing that this model of inactivity did not significantly alter the DMD zebrafish phenotype.

That being said, I appreciate the authors candor and understand the challenges both technical and personnel during the pandemic. I do believe that the revised manuscript does support the authors' reframed hypothesis(es) and I have no objections to the rest of the experiments.

*Reviewer #3:*

This study by Kilroy et al., is a revised version of the previously submitted manuscript. Authors have added more details to support their findings, however, mechanistic model still requires more studies to demonstrate a clear mechanism of NMES effect on skeletal muscle.

1. Does NMES result in formation of new myofibers or improves the attachment of existing myofibers? Are similar gene/pathways responsible for both?

2. Did authors look at the swimming behavior of fish during earlier time points (after NMES). Could a reduction in muscle degeneration could be due to low activity of fish during earlier time points following NMES treatment?

3. Figure 7D: Could differences observed in control dmd mutant and NMES treated mutant be due to variability in the initial damage (multiple somites in the control Vs isolated somites in NMES treated). As dmd phenotype is variable, have authors compared regeneration with other control dmd fish where muscle damage is not this profound?

4. Does hmox1a transcript expression correlate with the protein expression? Similarly, were changes were observed at the protein level for ECM genes identified by RNA-seq?

5. Failure of muscle improvement by eNMES in ITGA7 mutants may not mean that improvement in muscle structure in dmd mutant is through ITGA7. DMD mutant control muscle do have some expression of ITGA7 (Fontelonga et al., 2019 HMG) unlike ITGA7 null fish and therefore, failure of ITGA7 null fish to show improvement in NMES assay could be independent of DMD.

---

## [Author Response]

Essential revisions:1. The inactivity paradigm (e.g. figure 2) using tricaine as a means of inducing inactivity has pluses and minuses. There are issues with comparing it to rodent and human inactivity experiments (which usually involve hindlimb/limb immobilization), as the authors here are using chemical inhibition. Tricaine has systemic effects on multiple tissue types and organ systems including neurological and respiratory systems. I would be careful to call this model an inactivity model. Other appropriate models of inactivity exist, including physically restraining the zebrafish larvae to prevent movement and use of chemicals like BTS. In sum, the authors need to rule out if the consequences of tricaine administration is due to inactivity or pulmonary/secondary dystrophic pathology issues (e.g. swim bladder or respiration), and also consider a second "inactivity" paradigm in order to validate that the findings are due to inactivity.

We thank the reviewers for this critique because they are correct, tricaine does act on neural voltage-gated sodium channels in zebrafish. Indeed, the fact that tricaine does not paralyze muscle is advantageous to us given that we immobilize larvae in tricaine during electrostimulation. We were less concerned about pulmonary / swim bladder issues given that inactivity was applied at fairly young stages where circulation is not necessary for viability – but it is true that we did not rule this out.

The reviewers were prescient by highlighting that our data would be improved by incorporating an additional model of inactivity. We used the reviewers suggestion of BTS as an additional inactivity model and found that BTS does not exacerbate the DMD phenotype on its own. This result made us very grateful to the reviewers for their suggestion because we clearly need to delve further into why tricaine exacerbates the phenotype but BTS does not. For the purposes of clarity and focus, we have removed all inactivity data from this manuscript.

2. NMJ changes are hypothesized as an explanation for the response to NMES paradigms. Can/have the authors evaluated the functional output of the NMJ in the NMES-treated DMD zebrafish? Were any electrophysiological measurements performed on the NMES treated DMD fish, independent of any therapeutic experimental protocol?

We completely agree with the reviewers and would love to be able to conduct electrophysiology on zebrafish larvae. Unfortunately, this is an exceedingly difficult technique that is only done by a few labs and requires specialized equipment that we do not have access to (for reference, a pubmed search for zebrafish electrophysiology returned 240 papers – only 19 of which were neuromuscular recordings -the rest cardiac/brain/eye recordings). We were initially excited upon discovering a manuscript describing a set-up for undergraduate research labs (thinking that maybe we could do that), but this system only records the overall field potential of the startle response – which then opens up a host of confounding factors such as sensory physiology.

The reviewers are correct that this would be an excellent future direction – but this is likely one that would require a sabbatical because of the difficulty of this technique. We hope that the reviewers understand these constraints.

3. Hmox1 overexpression has been pursued as a strategy for DMD in mice by the Zoltan Arany and Joseph Dulak's groups, so the findings in figure 10 are supported. Have the authors evaluated whether or not the entire Hmox1 pathway was affected in the NMES-treated DMD fish?

The reviewers are correct in that Heme Oxygenase signaling has been identified as a potential strategy for DMD. We analyzed multiple members of the Hmox1 pathway and they were not differentially expressed and have included this analysis as a supplemental figure. We did however identify that HO is required for eNMES-mediated improvement (see response to comment #5).

4. For data presented in figure 1: authors describe the birefringence phenotype in mild mutants as increased degeneration for three days and then increased regeneration. Could they provide any experimental evidence of "muscle regeneration" mentioned in this statement?. Similarly, they mention severe dmd mutant regenerated throughout this study, however, no experimental data is provided to support this statement. As myotome contains both normal and degenerating myofibers, could improvement in birefringence be a consequence of the growth of those normal myofibers vs regeneration of sick myofibers? The term regeneration has also been used later in NEMS studies and needs to be supplemented with the experimental evidence of regeneration. In sum, there needs to be experimental support for the supposition of regeneration throughout the manuscript, as well as more careful consideration of rates of degeneration and regeneration.

We agree with the reviewers that we did not assess regeneration versus hypertrophy. We have removed these statements entirely and/or hedged by saying phases such as “improvement in muscle driven by regeneration and/or hypertrophy”.

5. There were several concerns with the transcriptomic data. More information related to the comparison of WT vs untreated sap is necessary for the interpretation of the changes seen with NMES. Also, the authors see very few consistent changes in terms of genes regulated one way in untreated sap and the other with treatment. It appears in fact that an overarching conclusion is that the transcriptional changes are NOT a driving force of the response to NMES. There is concern about the interpretability of taking one or two changed genes, as most transcriptional programs do not function in this isolated manner. For example, what is the true meaning of a change in b1 integrin without any changes in a integrin levels. In other words, how is this finding really of biological significance? In addition, the authors propose ECM changes as a potential mechanism. This should be supported by evaluation of ECM proteins (by western or immunostaining for example).

The reviewers are correct, and unfortunately GO analysis of the transcriptomic data was not particularly informative (as the reviewers surmised). In retrospect we likely should have conducted the transcriptomics experiments within a few hours of NMES rather than two days later. We did evaluate ECM proteins and unfortunately, at least in our hands, ECM staining of 7dpf DMD larvae is either extremely variable, or ECM distribution is extremely variable. Thus, we were unable to distinguish changes with eNMES because the baseline staining had so much variability.

Because of the limitations of the transcriptomic data, we now essentially gloss over the data as really only setting up two potential mechanisms that now test further in this revised manuscript: Cell adhesion and HO signaling.

6. The authors clearly demonstrate that the phenotype is variable. However, they do not carry this variability forward to all of their inactivity and exercise studies. Several of those studies feature relatively low n numbers. How have the authors ensured that the differences seen are not, in fact, due to the natural variability of the dmd phenotype in zebrafish? In particular, this seems like it would be an issue for all studies with n sizes less than 20, particularly given that the magnitude of difference for many of the studies is small.

We apologize for not being clear. We actually believe that the reviewers are correct – that the differences seen are due to the natural variability of the dmd phenotype, which we tried to define mathematically based on birefringence at 2 days post fertilization. We agree that the mild versus severe phenotype description distracts from the main narrative. Thus, we have removed that from this manuscript. In the future we hope to discern whether these phenotypes are transcriptionally distinct, use live imaging to determine rates of degeneration and regeneration and/or hypertrophy. Those experiments are outside the scope of this manuscript and will be the subject of future studies.

7. DMD is caused by damage in sarcolemma and subsequent myofiber detachment. The authors didn't observe any effect on myofiber structure but still found reduced velocity in mutants that were subjected to intermittent inactivity. Could this be due to a slight increase in sarcolemma damage (not examined here) and/or changes in the calcium in muscle fibers? Similarly, what are the effects of extended inactivity on MTJ structure? While authors make good observations with their animal model (as also seen in human and other animal models previously), mechanistic details underlying these changes are lacking.

We were unable to observe slight changes in membrane damage that would explain the change in swimming. One aspect of this manuscript that is interesting yet quite frustrating is that we rarely observe strict correlations between muscle structure, NMJ structure, swimming, and survival. We do not know why these data are so confusing, but we suspect that we will need to develop more sophisticated assays to get at underlying mechanisms in the future.

[Editors’ note: further revisions were suggested prior to acceptance, as described below.]

The manuscript has been improved but there are some remaining issues that need to be addressed, as outlined below:There are three major areas of concern related to the resubmitted manuscript.(1) The authors fail to truly contextualize their findings with the abundant literature related to neuromuscular electrical stimulation in neuromuscular disease. This needs to be incorporated into the Discussion section in order to best interpret the current study within the field. Please see the comments from reviewer 1 below.

We have added much more context for NMES and disease. We hope that the reviewers appreciate our approach to broadening the discussion. The added text :

“Electrical stimulation has been shown to be generally safe and potentially effective for some conditions. For example, there are potential therapeutic benefits of NMES for treatment of spinal cord injuries. Although not all trials observe an increase in voluntary muscle strength with NMES, none found deleterious effects of NMES (de Freitas et al., 2018). NMES can also improve dysphagia after stroke: ten out of 11 trials showed that NMES improved swallowing with only one showing no effect (Alamer et al., 2020). NMES combats disuse atrophy in multiple contexts. Chronic NMES applied to mice who were anesthetized for 2.5 weeks not only showed increased muscle mass in the stimulated limb, but also had improved insulin sensitivity (Lotri-Koffi et al., 2019). NMES for at least seven days is sufficient to improve muscle mass of lower limbs in non-ambulatory patients with traumatic brain injury (Silva et al., 2019). NMES is even being studied as a means to combat muscle atrophy during spaceflight (Maffiuletti et al., 2019). The above data show that NMES has potential benefits for sudden muscle disuse caused by external events, but the molecular and cellular mechanisms are not well understood.

NMES also shows promise for neurodegenerative disorders and aging muscle. Muscle mass and strength are improved in aged rats with NMES and NMES improves muscle mass and balance in older adults as well as older adults with dementia (Dow et al., 2005; Kern et al., 2014; Nishikawa et al., 2021). NMES improves mobility in patients with progressive multiple sclerosis (Wahls et al., 2010). NMES may also improve mobility and strength in ALS (Handa et al., 1995), although intensity may be important (The ALSUntangled Group, 2017). Far less is known about NMES in the context of muscular dystrophies. The concept of super-imposing electrical stimulation to improve dystrophic muscle was proposed by the neurologist who first described DMD over a hundred years ago (Duchenne, GB, 1870). Despite the longevity of this hypothesis, it has not been sufficiently tested as a therapy for DMD. There are promising data: NMES improves muscle fiber morphology in dystrophic mice (Dangain and Vrbova, 1989; Luthert et al., 1980; Vrbová and Ward, 1981) and chickens (Barnard et al., 1986). In chickens the benefit was most pronounced if administered prior to rampant muscle degeneration (Barnard et al., 1986). Small trials in young children also suggest that early low frequency NMES can improve voluntary muscle contraction compared with the contralateral leg (Scott et al., 1990, 1986). NMES can also improve muscle function in myotonic dystrophy (Chisari et al., 2013) and limb girdle muscular dystrophy (Kılınç et al., 2015). Despite these promising studies, NMES is not commonly used as an adjuvant therapy in myopathies and dystrophies. This is potentially due to the impractical approach of chronic NMES for most if not all skeletal muscles. Thus, it is important to elucidate underlying molecular and cellular mechanisms of beneficial impacts of NMES. It is known that electrical stimulation increases both the number and size of AChR clusters in primary myoblasts. The fact that expression of Rhapsyn is also increased indicates that the increased AChR clusters are leading to increased mature NMJs. However, clearly more mechanistic studies regarding the effects of NMES on muscular dystrophies are warranted.”

(2) The authors suggest that NMES may be working through modification of ECM-cell adhesion. They present data showing failure of itga7 mutants to respond to NMES. While this provides evidence that itga7 is potentially involved in mediated response in WT fish, it does not inform on the situation with dmd. The ideal experiment would of course be to test NMES in the setting of itag7/dmd double knockouts. At the very least, the fact that this link is not firmly established in the dmd model needs to be pointed out.

We agree with the reviewers that we only tested itga7 in the context of itga7 mutant fish. We have not yet generated double mutants, mostly because data from Szatl et al., 2012 (Epistatic dissection of laminin-receptor interactions) strongly suggests that itga7;dmd double mutants would be so disrupted that it would be impossible to interpret results (The Currie group generated dmd;integrin linked kinase double mutants and showed severe disruption).

We have added the following sentences:

– To the Results section: “Thus, Itga7 is required for eNMES-mediated improvement of muscle structure, at least in the context of itga7 mutants”.

– To the Discussion section “Itga7 mutant larvae did not improve with eNMES, indicating that Itga7 is required for eNMES-mediated improvement. We did not generate dmd;itag7 double mutants to test whether eNMES is not effective in this context, which would be interesting to do.”

(3) The authors also implicate TGFb signalling based on RNAseq data. The authors rightly point out the short comings of RNAseq for uncovering the important pathways; this is particularly true for TGFb signalling, which is very much regulated and governed at the level of post translational changes. There are simple assays for examining TGFb signalling and activity that have been utilized in other DMD models (such as mdx) and in zebrafish. Such assays would lend support to the RNAseq data, which on its own its relatively weak proof of the involvement of the ECM and TGFb.

The reviewers are correct that Tgfbeta-induced is downstream of TGFb signaling, and that RNAseq data is not strong evidence for the role of cell adhesion to the ECM changing in response to neuromuscular electrical stimulation. This is why we conducted the cell adhesion experiments shown in Figure 9, which suggest that cell adhesion is increased with eNMES.

We agree with the reviewers that using an assay for TGFb signaling at the protein level would be more informative. We thus stained control and eNMES wild-type and control and eNMES mutants for p-Smad. Please note that this has not previously been done in older fish. We did the experiment twice and imaged 68 embryos at at least three locations – anterior, middle, and posterior at 5, 6, and 7 days old. Unfortunately, the only conclusion that we can make from these data is that we can’t conclude anything. p-Smad staining is incredibly variable in zebrafish skeletal muscle at 5,6,7 days (this has not previously been investigated). As an example I put one panel (Author response image 1) that highlights just a couple of the muscle fibers with p-Smad positive nuclei in a wild-type embryo at 7 days (yellow arrows point to positive nuclei), as well as quantification of the data at 5 days – clearly showing dramatic variability. We suspect that TGFb signaling at this time point is dynamic, and thus capturing consistent changes in p-Smad readout in fixed larvae is impossible.

**Author response image 1. sa2fig1:** 

Developing tools and image quantification methods for live readouts of TGFb signaling is an interesting future direction but out of the scope of this manuscript. We hope the reviewers agree with us that the cell adhesion assay clearly implicates cell adhesion, although the underlying mechanisms are not yet identified.